# Clinically relevant molecular subtypes and genomic alteration-independent differentiation in gynecologic carcinosarcoma

Osamu Gotoh [1], Yuko Sugiyama[2], Yutaka Takazawa[3], Kazuyoshi Kato[2], Norio Tanaka [1], Kohei Omatsu[2], Nobuhiro Takeshima[2], Hidetaka Nomura[2], Kosei Hasegawa[4], Keiichi Fujiwara[4], Mana Taki[5], Noriomi Matsumura[5], Tetsuo Noda[1] & Seiichi Mori[1]*

Carcinosarcoma (CS) of the uterus or ovary is a rare, aggressive and biphasic neoplasm composed of carcinoma and sarcoma elements. Previous genomic studies have identified the driver genes and genomic properties associated with CS. However, there is still no molecular subtyping scheme with clinical relevance for this disease. Here, we sequence 109 CS samples, focusing on 596 genes. We identify four molecular subtypes that resemble those observed in endometrial carcinoma: *POLE*-mutated, microsatellite instability, copy number high, and copy number low subtypes. These molecular subtypes are linked with DNA repair deficiencies, potential therapeutic strategies, and multiple clinicopathological features, including patient outcomes. Multi-regional comparative sequencing reveals genomic alteration-independent CS cell differentiation. Transcriptome and DNA methylome analyses confirm epithelial-mesenchymal transition as a mechanism of sarcoma differentiation. The current study thus provides therapeutic possibilities for CS as well as clues to understanding the molecular histogenic mechanism of its development.

---

[1] Project for Development of Innovative Research on Cancer Therapeutics, Cancer Precision Medicine Center, Japanese Foundation for Cancer Research, 3-8-31 Ariake, Koto-ku, Tokyo, Japan. [2] Department of Gynecology, Cancer Institute Hospital, Japanese Foundation for Cancer Research, 3-8-31 Ariake, Koto-ku, Tokyo, Japan. [3] Department of Pathology, Cancer Institute Hospital, Japanese Foundation for Cancer Research, 3-8-31 Ariake, Koto-ku, Tokyo, Japan. [4] Department of Gynecologic Oncology, Saitama Medical University International Medical Center, 1397-1 Yamane, Hidaka-shi, Saitama, Japan. [5] Department of Gynecologic Oncology, Kyoto University Hospital, 54 Kawaharacho, Shogoin, Sakyo-ku, Kyoto, Japan. *email: seiichi.mori@jfcr.or.jp

Carcinosarcoma (CS) of Mullerian origin (previously called malignant mixed Mullerian tumor) is defined as a biphasic, malignant tumor composed of both carcinoma and sarcoma elements that arises in the uterus, ovary, Fallopian tube, and peritoneum[1–5]. Histologically, the epithelial component can be endometrioid, serous, undifferentiated, clear cell, or other type and is usually observed as endometrial or ovarian carcinoma. Sarcomas are classified as homologous and heterologous, depending upon whether the tissue is normally present in the uterus[5–7]. The nature of CS is more similar to that of its carcinoma counterpart presenting with the same histology rather than that of sarcoma counterpart in terms of risk factor profile, dissemination pattern, and sensitivity to cytotoxic agents. Consequently, CS is generally considered a derivative of carcinoma and is therefore treated with the same therapeutic strategy[1,2].

Three major theories have been proposed to explain the histogenic process of CS: collision, combination, and conversion theories. The collision theory suggests that synchronous but independent carcinoma and sarcoma coincidently arise and merge. The combination and conversion theories assume that the two components are derived from a common precursor with divergent and metaplastic differentiation, respectively[1,5]. Although recent evidence suggests a monoclonal origin for CS, based on various marker expression or genetic alterations[3,8–14], it remains largely unclear whether divergence or metaplasticity is the dominant histogenic mechanism.

Despite the considerable similarity between CS and its carcinoma counterpart, previous epidemiological studies have shown that CS is associated with poorer outcomes[2,3]. Not only is CS more frequently diagnosed at a more advanced stage but patients also exhibit worse survival rates, even after patient stratification by histology, stage, or primary site, indicative of a more aggressive disease[2,15]. Furthermore, even though several clinicopathological assessments have identified multiple independent prognostic factors for CS[7,15], the molecular basis of the aggressiveness of this disease is still largely unknown. Hence, there is a clinical demand to discover actionable molecular targets as well as biomarkers for prognostication and patient stratification.

Previous studies have linked various types of genomic instability, such as microsatellite instability and copy number aberrations, with the clinicopathological features of cancer, as exemplified by tumor histology, stage, and patient prognosis. Genomic instability profiles have been used for the molecular subtyping of tumor samples in many cancer types[16]. Microsatellite instability, caused by defective mismatch repair, is observed in 20–40% of endometrial endometrioid carcinoma and is associated with higher histological grade and more lymphovascular space invasion but better prognosis[17]. The ultramutated phenotype, resulting from mutations in the exonuclease domain of DNA polymerase ε (POLE), is present in several cancer types, including endometrial cancer, and linked with diseases at an earlier stage and with a more favorable prognosis[18]. Copy number aberration is a dominant characteristic of the genome in endometrial and ovarian serous carcinoma, and is associated with a more advanced disease and poor patient outcomes[16,19].

Recent work evaluating CS genomic instability profiles has shown that microsatellite instability accounts for 5–21% of gynecologic CS[12,20–23]. Other studies have described a few CS cases with the ultramutated defective POLE phenotype among genomic cohorts[24–26]. Through arrays of comparative genomic hybridization, several groups have revealed aberrant patterns within the CS genome and identified similarities between CS and serous carcinoma[27,28]. Despite these previous efforts and other recent comprehensive genomic studies[14,24,26,29], the genomic deregulation pattern among patients with CS and its contribution to the aforementioned clinicopathological properties remain to be elucidated.

Here, we present the results obtained through targeted massive parallel sequencing analyses as well as DNA methylome and transcriptome analyses of uterine and ovarian CS samples. Based on genomic profiling data, we identify four molecular subtypes that not only correlate with various clinicopathological features but provide options for potential therapeutic strategies. Clonal origin and the retention of driver events in carcinoma and sarcoma elements from single tumor samples are confirmed by differential sequencing of both components. Further multiregional exome sequencing reveals differentiation of the carcinoma and sarcoma components is independent of genomic alterations. Moreover, we show that sarcoma differentiation is associated with epithelial–mesenchymal transition (EMT)-related gene expression and DNA methylation changes. These findings support the histogenic hypothesis of the clonal origin of CS with metaplastic conversion.

## Results

**Identification of four genomic subtypes in CS samples.** We used a modified version of the molecular subtyping method based on genomic aberration profiling[16] (described in the Methods) on 109 uterine and ovarian CS samples, and then used a decision tree to classify samples into four molecular subtypes: POLE-mutated (POLE), microsatellite instability (MSI), copy number high (CNH), and copy number low (CNL) subtypes. POLE is designated by a somatic POLE mutation in the exonuclease domain, and is characterized by a substantial number of SNVs (range: 190–1999, median: 479) with a variable number of indels (range: 1–56, median: 3) and CNVs (range: 0–28, median: 1). CS samples with high microsatellite instability (MSI-H) but not POLE proofreading domain mutations were classified as MSI tumors. MSI tumors exhibited numerous indels (range: 10–63, median: 27) and a moderately increased number of SNVs (range: 8–207, median: 55) but fewer CNVs (range: 0–7, median: 2). After POLE and MSI assignment, we performed an unsupervised hierarchical clustering analysis using DNA copy number data. Samples were classified as CNH subtype when they showed highly aberrant copy number alterations (range: 1–48, median: 17) resembling the cluster 4 in the uterine TCGA (The Cancer Genome Atlas) analysis[16]. CNH subtype samples exhibited a minimal number of alterations in SNVs (range: 1–19, median: 6) and indels (range: 0–6, median: 1). The remaining tumors were assigned as CNL subtype, with few SNVs, indels, or CNVs (Fig. 1). The patterns of nucleotide substitution differed among the subtypes: the T to C transition and T to G transversion predominantly occurred in MSI and POLE subtypes, respectively. The CNH subtype was tightly linked with high tumor ploidy ($p = 0.0002$; Mann–Whitney $U$-test). The genomic aberration subtypes were unrelated to tumor purity (Fig. 1a).

Histologically, the carcinoma component of most POLE and MSI tumors was endometrioid, whereas tumors with serous histology were mainly classified into the CNH subtype (Fig. 1 and Supplementary Fig. 3A). The distribution of sarcoma histology was not biased among the subtypes (Fig. 1). Fifteen of the 17 ovarian CS samples exhibited a CNH phenotype (Fig. 1 and Supplementary Fig. 3A). One ovarian CS (OV594), classified as POLE, was associated with an endometriotic lesion in the ovary (grade-1 endometrioid carcinoma) that was histologically distinct from the carcinoma component (grade-2 endometrioid). In accordance with previous criteria, the case was diagnosed as synchronous endometrial and ovarian carcinoma (SEOC). The ovarian CS and endometrial carcinoma share a subset of SNVs/indels, including the POLE p.P286R mutation. In view of this

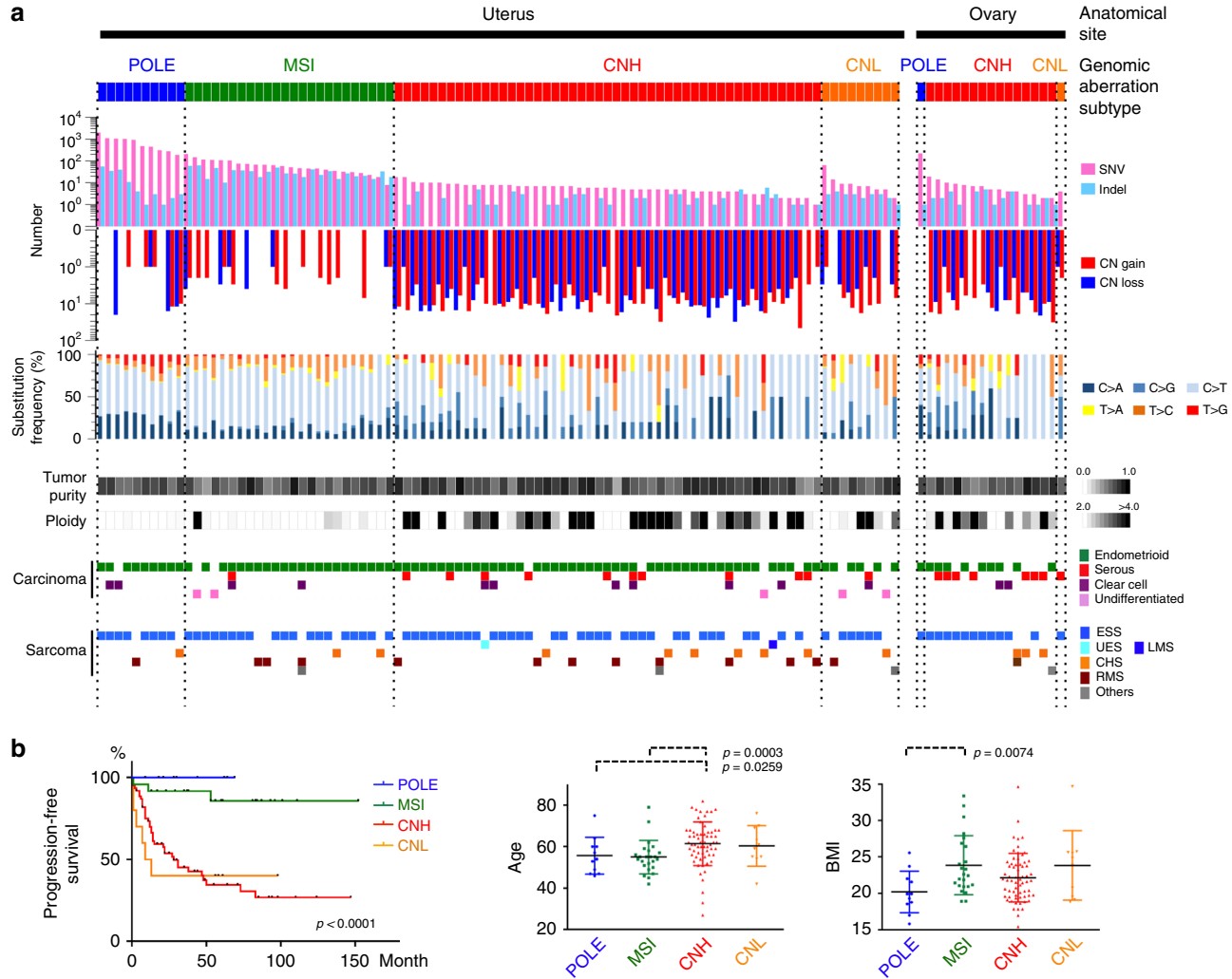

**Fig. 1** Genomic aberration profiling identified four molecular subtypes that correlate with various clinicopathological features. **a** Patterns of genomic alterations in carcinosarcoma (CS) genomic aberration subtypes (MSI, POLE, CNH, and CNL). Uterine (92) and ovarian (17) samples were sorted according to the number of SNVs within a subtype. Panels from top to bottom: bar plots for number of SNVs (pink) and indels (pale blue); bar plots for number of segments with copy-number aberration (CN gain, red; CN loss, blue); rates in percent of nucleotide substitutions (C > A, indigo; C > G, navy blue; C > T, sky blue; T > A, yellow; T > C, orange; T > G, vermilion); tumor purity; tumor ploidy; histopathological diagnosis of carcinoma components (endometrioid, green; serous, red; clear cell, violet; undifferentiated, pink); and histopathological diagnosis of sarcoma components (ESS, sky blue; RMS, brown; CHS, yellowish brown; others, gray). POLE, *POLE*-mutated; MSI, microsatellite instability; CNH, copy number high; CNL, copy number low; SNV, single nucleotide variant; CNV, copy number variant; CN, copy number; ESS, endometrial stromal sarcoma; RMS, rhabdomyosarcoma; CHS, chondrosarcoma. **b** Clinical relevance of CS genomic aberration subtypes. Statistically significant parameters are shown among CS genomic subtypes. Left panel. Relapse-free survival in Kaplan–Meier curve. *P*-values were computed with log-rank test. Middle panel. Age. *P*-values were computed with the Mann–Whitney *U*-test. Right panel. Body mass index (BMI). *P*-values were computed with the Mann–Whitney *U*-test. Bars and error bars indicate mean and standard deviation

association, we concluded that this ovarian POLE CS of OV594 likely originates from the endometria and shares ancestry with the synchronous endometrial carcinoma (see Supplementary Table 1 and Supplementary Fig. 4).

Statistical analyses with clinical parameters revealed significant correlations for POLE, MSI, CNH, and CNL subtypes with patient outcomes in terms of progression-free survival and overall survival (Fig. 1b; POLE: $p = 0.9960$ and 0.9966, HR = 0.0000 and 0.0000; MSI: $p = 0.0017$ and 0.0099, HR = 0.1530 and 0.0728; CNH: $p = 0.0001$ and 0.0074, HR = 4.1475 and 2.9268; CNL: $p = 0.2107$ and 0.0053, HR = 1.7278 and 3.2571 by univariate Cox regression model). These strong correlations between genomic aberration subtype and patient survival were still observed after excluding the ovarian data (POLE: $p = 0.9964$ and 0.9969, HR = 0.0000 and 0.0000; MSI: $p = 0.0027$

and 0.0133, HR = 0.1629 and 0.0801; CNH: $p = 0.0002$ and 0.0110, HR = 4.4282 and 3.0380; CNL: $p = 0.2612$ and 0.0064, HR = 1.7176 and 3.5378 by univariate Cox regression model); whereas the ovarian CS exhibited a more unfavorable prognosis than CS originating in the uterus (Supplementary Fig. 1A). Moreover, the multivariate Cox regression analysis with FIGO stage, tumor size, and primary tumor anatomical site confirmed that each of these genomic aberration subtypes was independent (POLE: $p = 0.9985$ and 0.9987, HR = 0.0000 and 0.0000; MSI: $p = 0.0068$ and 0.0205, HR = 0.1937 and 0.0937; CNH: $p = 0.0007$ and 0.0296, HR = 3.9048 and 2.6047; CNL: $p = 0.1782$ and 0.0025, HR = 1.9214 and 4.1512). Age at diagnosis was highest in patients with the CNH subtype, and body mass index (BMI) measures were lowest for patients with the POLE subtype, as compared with the other subtypes (Fig. 1b). Such

associations imply distinct biological properties for each of the four genomic subtypes.

To determine the reproducibility of this molecular subtyping method in an independent cohort, we applied the same scheme to data derived from 57 TCGA uterine CS samples[26]. The analysis reproduced the same four subtypes; however, the proportions in each subtype were significantly different from those in our cohort: 1 (1.8%) POLE, 2 (3.6%) MSI, 50 (91.0%) CNH, and 2 (3.6%) CNL (Supplementary Fig. 3A; $p < 0.0001$ by Fisher exact test).

**Genomic instability and DNA repair defects in CS subtypes.** The genomic aberration profiles observed among the CS subtypes prompted us to investigate the relationship between CS subtype and DNA repair defects. We curated gene sets for mismatch repair (MMR) and homologous recombination (HR) pathways that were relevant for hereditary breast, endometrial, and ovarian cancer[30], and subsequently assessed whether a tumor retained a germline and/or somatic mutation or CpG-site hypermethylation

in the selected genes[31]. The results are shown together with the somatic mutational status of POLE and TP53 (Fig. 2). All tumors of the POLE subtype exhibited somatic mutation of the POLE exonuclease domain, as designated (100%; 11 of 11 cases). Defective MMR was observed in the MSI subtype, including MLH1 promoter hypermethylation and germline/somatic mutations in MLH1, MSH2, MSH6, or PMS2 loci. HR deficiency was a distinctive feature of the CNH subtype, as exemplified by (1) BRCA1 and RAD51C promoter hypermethylation; (2) germline inactivation plus loss of heterozygosity (LOH) in the loci of BRCA1/2, ATM, RAD50, and BLM; and (3) somatic mutation of the PTEN gene[31]. Most cases (22/24; 91.7%) in the MSI subtype and about one-third (23/64; 35.9%) of the CNH tumors exhibited defects in MMR and HR pathways, respectively. Of note, these genetic and epigenetic changes were observed in samples within a genomic aberration subtype in a mutually exclusive manner.

CCNE1 amplification and homozygous deletions of RB1 and NF1 loci were previously reported to be involved in chromosomal instability through a distinct mechanism not involving HR deficiency (non-HRD)[32]. Within the CNH subtype, 17 (26.6%) of

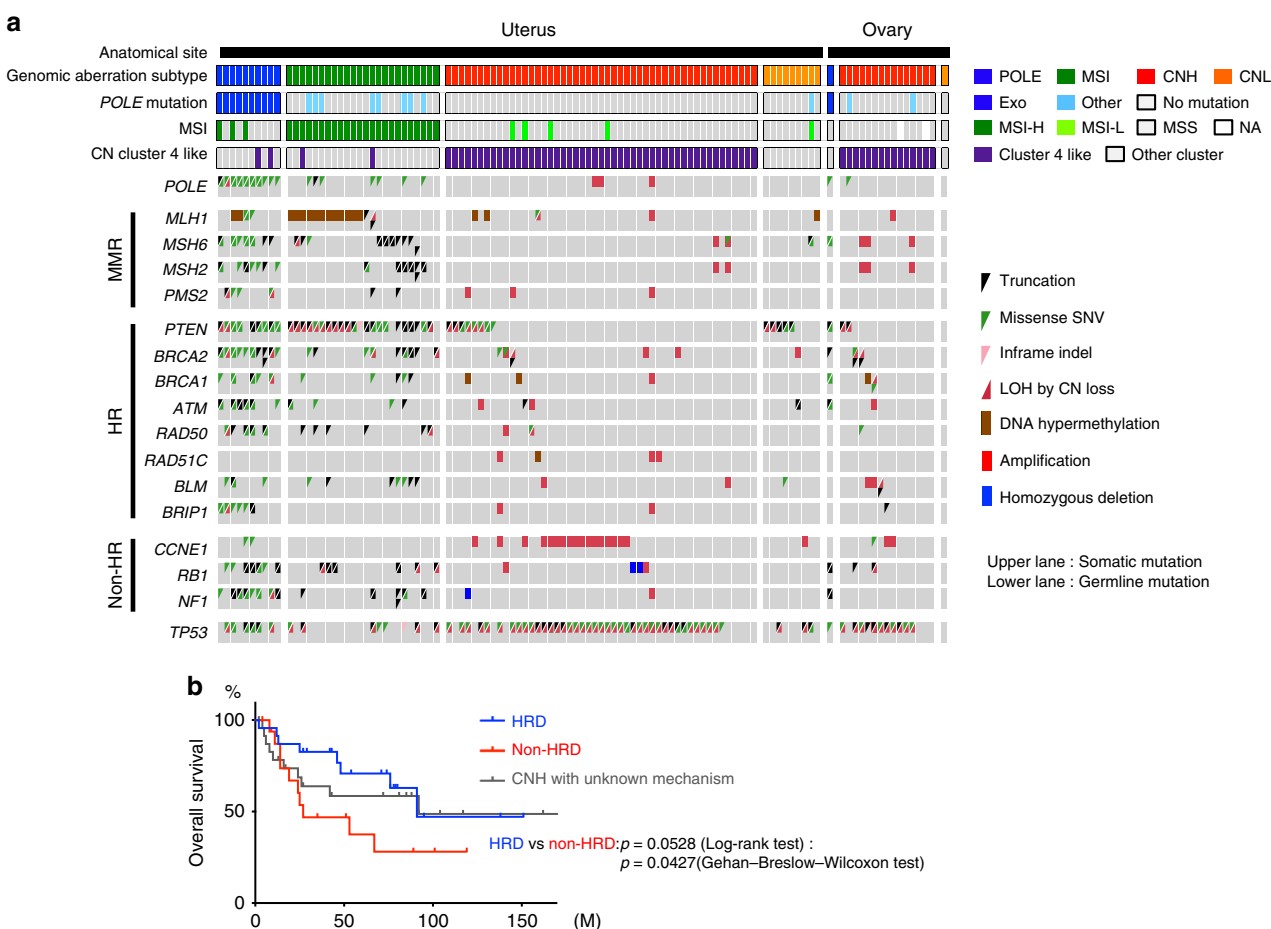

**Fig. 2** Germline and somatic inactivation of DNA repair genes in carcinosarcoma (CS) genomic aberration subtypes. **a** Relationship between genomic aberration subtype and DNA repair gene mutation. Status of inactivation of POLE, mismatch repair (MMR), and homologous recombination (HR) genes are sorted according to CS subtype and shown along with mutational status of TP53 using Oncoprint. Color code is as follows: truncating mutation, black triangle; missense SNV, green triangle; in-frame indel, pink triangle; loss of heterozygosity (LOH) in wildtype allele by copy-number loss, red triangle; promoter hypermethylation, brown rectangle; amplification, red rectangle; and homozygous deletion, blue rectangle. Somatic and germline mutations are shown in the upper and lower rows, respectively. Using Oncoprint, we show germline missense SNVs only when pathogenic or likely pathogenic, as according to the American College of Medical Genetics and Genomics, Association for Molecular Pathology (ACMG) guidelines (see Methods). Note that 2 of 24 MSI cases (one had MLH1 c.306 + 1_306 + 2delGT, another had MSH2 p.Q374X and MSH6 c.4001 + 2_4001 + 5delTAAC), and 5 of 64 CNH cases (BRCA1 c.5278-1G > A [this case was reported in Abe 2014[68]], BRCA2 p.V1447X, BRCA2 p.T1858fs, BRCA2 p.Y2154fs, and BLM p.K1056fs) had germline mutations, all of which had lost the wildtype allele by LOH or by additional somatic truncation in the tumor samples. **b** Overall survival of patients with CNH tumors with HRD, non-HRD and the unknown mechanisms. HRD: Homologous recombination deficiency

64 CNH tumors (15 *CCNE1* amplification and 2 *RB1* deletion) exhibited this non-HRD-type chromosomal instability. The remaining 24 CNH samples had no apparent genomic features. The CNH subtype with non-HRD chromosomal instability showed worse prognosis than CNH tumors with HRD in terms of overall survival ($p = 0.0427$ by Gehan-Breslow-Wilcoxon test; Fig. 2b). The same trend was also detected after excluding the ovarian CS patient data ($p = 0.1197$ by Gehan-Breslow-Wilcoxon test). Collectively, these findings demonstrate the relevance of the background genotype or epigenotype in causing the CNH phenotype.

**Driver gene mutations in CS.** To identify driver mutations in the present CS cohort, we performed statistical analyses with IntO-Gen software (Oncodrive-fm and OncodriveCLUST) on somatic SNV/indels (focusing on 596 target panel genes; $n = 109$) and with GISTIC on CNVs (detected by SNP6 arrays; $n = 105$)[33,34]. From these analyses, we identified 40 genes and 10 copy number segments as CS driver events (Fig. 3 and Supplementary Fig. 5).

For SNVs and indels, *TP53*, *PTEN*, *ARID1A*, *PIK3R1*, *CTCF*, *PPP2R1A*, *RPL22*, *INPPL1*, *MSH2*, and *KRAS* were highly significantly mutated ($q < 0.00001$). Of these, seven drivers (*TP53*, *PTEN*, *ARID1A*, *PIK3R1*, *PPP2R1A*, *MSH2*, and *KRAS*) were included among the highly significantly mutated genes from the TCGA UCEC dataset[16]. The extent of overlap was similar to that of previous CS genomic cohorts[13,14,24,26]. Also, by GISTIC analysis of SNP6 data, copy number amplification was highly significant for *MECOM*, *MYC*, and *CCNE1*. Because of the hypermutator phenotype, 33 (83%) and 23 (58%) of the 40 SNV/indel driver genes were, respectively, super-recurrently mutated and exhibited bias in the distribution to POLE and MSI subtypes ($p < 0.05$, Fisher exact test). Other than such driver enrichment in hypermutators, only *TP53* was identified as a CNH-enriched driver mutation ($p < 0.0001$, Fisher exact test; Supplementary Fig. 5A). Among CNV driver events, *CCNE1* and *MECOM* amplifications were significantly enriched in the CNH subtype ($p = 0.0042$ and $p = 0.01452$, Fisher exact test; Supplementary Fig. 5B).

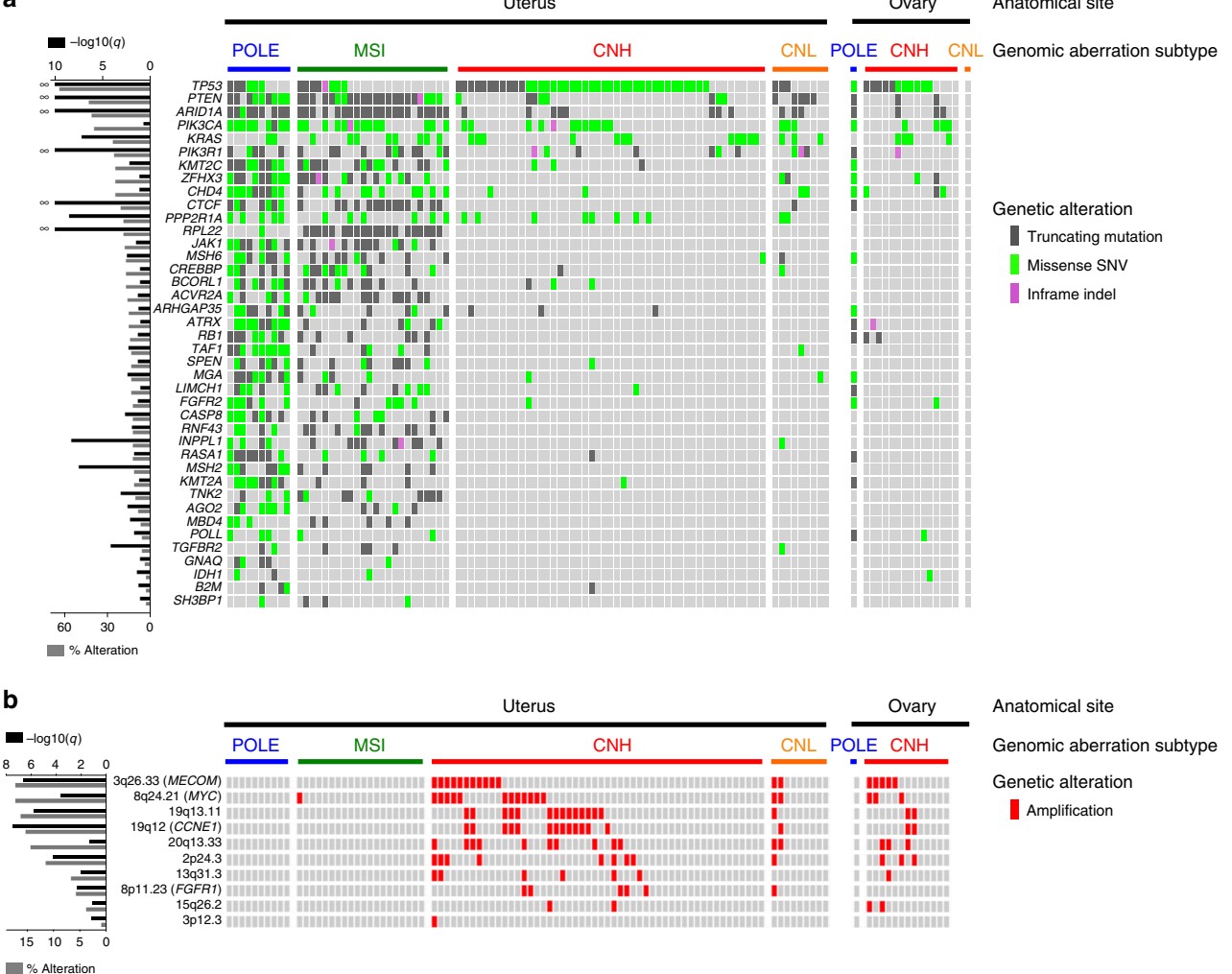

**Fig. 3** Somatic driver genes in carcinosarcoma (CS) subtypes. **a** Somatic driver SNVs and indels. Driver genes were computed by IntOGen[54] using somatic SNVs and indels (from target panel or exome sequencing data focusing on 596 genes of the target panel; $n = 109$). The mutations are presented with Oncoprint according to CS genomic subtype in descending order of mutational frequency in the cohort. Driver genes with statistical significance of $q < 0.02$ are shown. Mutational frequency in the cohort (gray bar) and $q$-value for each driver gene (black bar) are shown on the left. Color code is as follows: somatic missense SNV, yellow green; somatic truncating mutation, gray; and somatic in-frame indel, lilac. **b** Somatic driver CNVs. CNV driver genes were detected by GISTIC[34] using CNVs called (from SNP6 array data; $n = 105$)[52]. Mutational frequency in the cohort (gray bar) and $q$-value for each driver gene (black bar) are shown on the left. Amplification (CN ≥ 6) is shown in red[53]. We do not show copy number gain (CN = 3, 4, or 5) or loss (CN = 1). Note that only amplification was detected, and no homozygous deletion (CN = 0). CN; copy number

Across the 40 SNV/indel driver genes, 9 (23%) and 22 (55%) genes were identified as significantly correlated with better prognosis in terms of overall survival and progression-free survival, respectively, as determined using a univariate Cox regression analysis. However, in the multivariate analyses, all of these genes were identified as confounding with the POLE, MSI, or both subtype assignment. Through further prognosis-correlation analyses within each subtype, we identified *ARID1A* as the only gene correlated with poor outcomes in the CNH subtype (Cox $p = 0.0114$, HR = 2.7880; log-rank test $p = 0.0075$ in progression-free survival) (Supplementary Fig. 6).

Recently, a simple classification method was proposed, based on the mutational status of the major drivers for endometrioid (*PTEN* and *ARID1A*) and serous (*TP53* and *PPP2R1A*) histotypes[22,26]. In this method, endometrial cancer is subdivided into endometrioid- and serous-like genotypes (or mutational subtypes)[22,26]. Whereas most (84.9%; 158/186 cases) of the TCGA uterine endometrioid carcinomas showed an endometroid-like mutational pattern, CS samples with an endometrioid carcinoma component from both TCGA and the current (herein designated as Japanese Foundation for Cancer Research; JFCR) cohorts had some proportion of the serous-like subtype. Conversely, the serous-like genotype coincided with serous histology of TCGA uterine carcinomas, as well as CS samples from the TCGA and JFCR cohorts (Supplementary Fig. 3B).

**Molecular features of genomic aberration subtype.** The molecular subtypes identified by genomic aberration patterns and transcriptomic/epigenetic subtyping were coincident, which implied that distinctive features for each molecular subtype should also be detectable by other assay methods (Supplementary Note 2, Supplementary Figs. 7 and 8). Previous research into uterine CS by The Cancer Genome Atlas (TCGA UCS) uncovered molecular similarity among most CS samples with the CNH subtype of endometrial cancer and serous ovarian cancer in terms of copy number, DNA methylation, and mRNA expression data[26]. Thus, to distinguish serous and endometrioid histology, we first used the TCGA uterine corpus endometrial cancer (TCGA UCEC) dataset to generate expression and DNA methylome signatures. As anticipated, the signature subdivided the training TCGA UCEC data into two subclasses: the CNH subtype and other subtypes. The number of significantly aberrated segments derived from copy number analysis with GISTIC also showed the distinct nature of the CNH subtype compared with the other subtypes. As described in the TCGA UCS study[26], the driver mutation-based genotypes and the serous-like and endometrioid-like genotypes coincided with CNH and the other subtypes of TCGA UCEC data (Fig. 4). When the same signatures were applied to the genomic aberration subtypes in the current cohort, particularly for uterine CS samples (JFCR UCS), we identified an equally distinctive pattern in the omics data. Importantly, the same features were also detected in the genomic

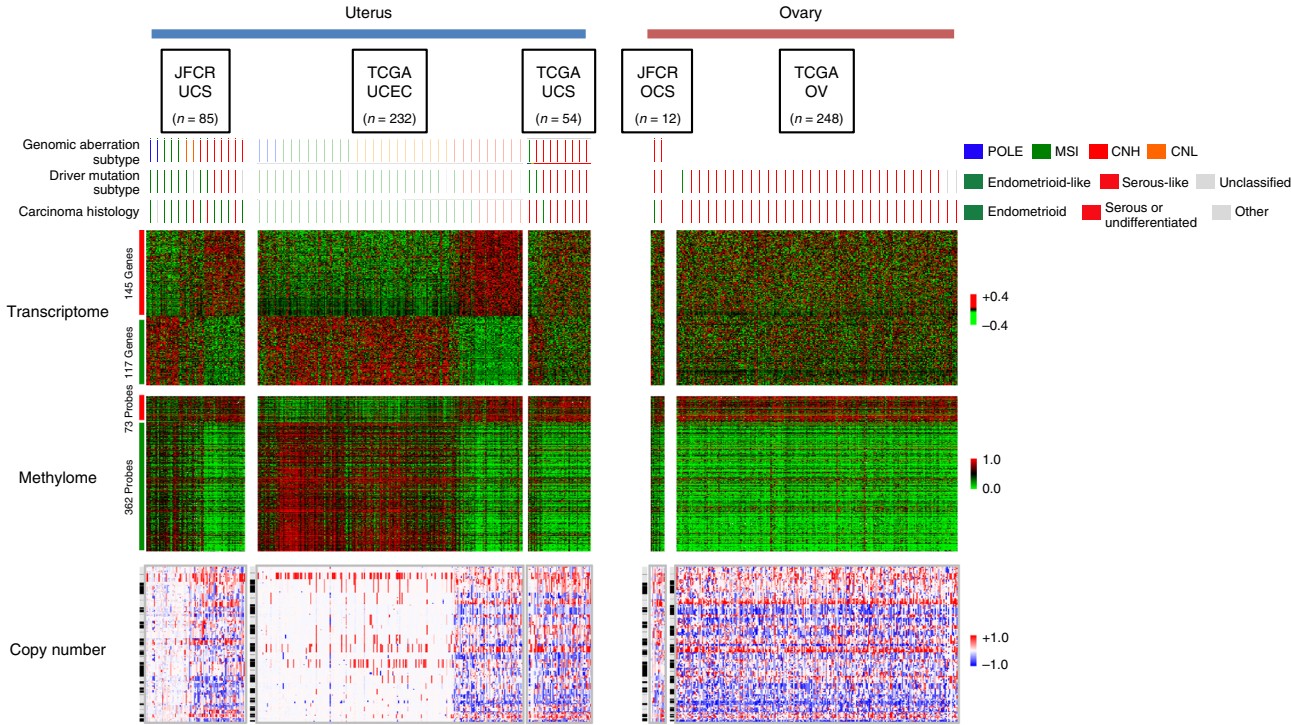

**Fig. 4** Genomic aberration subtypes in the transcriptome, DNA methylome, and copy number data derived from uterine and ovarian cancers. Heatmaps for gene expression (upper panel; red and green color indicate high and low expression), CpG methylation (middle panel; red and green color indicate hyper- and hypo- methylation of the CpG site) and aberrant copy number segments (lower panel; red and blue color indicate gain and loss of segments in copy number) are shown. For the transcriptome or DNA methylome, serous (red line) vs. endometrioid (green line) signatures were generated using the uterine corpus endometrial carcinoma datasets in The Cancer Genome Atlas repository (TCGA UCEC $n = 232$) and applied to uterine and ovarian carcinosarcoma (UCS and OCS) datasets from the JFCR (Japanese Foundation for Cancer Research), TCGA (JFCR UCS $n = 85$, TCGA UCS $n = 54$, and JFCR OCS $n = 12$), and from TCGA high-grade serous ovarian cancer (TCGA OV $n = 248$). Data are presented as heatmaps. The serous vs. endometrioid signatures consisted of 145 and 117 genes for the transcriptome and 73 and 362 probes for the DNA methylome. For copy number data, the heatmaps show abnormal segments detected by EXCAVATOR and GISTIC in chromosomal order. Genomic aberration subtypes (POLE, blue; MSI, green; CNH, red; and CNL, orange), driver mutation subtypes (endometrioid-like, green; serous-like, red; and unclassified, gray), and carcinoma histology (endometrioid, green; serous, red; and the other histologies, gray) are presented above the panels

aberration subtypes within the TCGA UCS dataset despite the differences in the proportions of the subtypes (Fig. 4). It is also of note that the transcriptome pattern of JFCR ovarian CSs (JFCR OCS) better resembled that of the TCGA high-grade serous ovarian carcinomas (TCGA OV) than those of the TCGA UCEC or TCGA UCS, implying a distinction in the nature of uterine and ovarian CSs (Fig. 4).

**Evolution of carcinoma and sarcoma components.** Previous molecular marker-based studies have highlighted the clonal nature of carcinoma and sarcoma elements in a CS tumor[3,8–14]. However, clonal evolution has not been fully elucidated, particularly in the context of genomic aberration subtype and inter-tumor heterogeneity across individual samples. To evaluate this, phylogenetic trees were constructed based on similarity of SNVs and indels using a 596-gene panel. These tested samples were taken from tissues containing both carcinoma and sarcoma components situated adjacently and captured discretely by laser-capture microdissection. Samples from 9 cases (GY141, GY165, DKK008, GY147, GY140, DKK017, EN333, GY146, and OV507) and 5 cases (EN325, GY069, GY087, GY127, and GY129) were examined with the 596-gene panel and exome sequencing, respectively. The trees are presented along with the timings of driver mutation acquisitions using the cancer cell fraction (CCF), as computed from mutant allele frequencies (Fig. 5 and Supplementary Table 4). Trunks and branches represent shared and private mutations, respectively, with the length proportional to the number of SNVs and indels. We compared the evolutionary paths of the carcinoma and sarcoma components from 14 primary tumors across the four genomic aberration subtypes (2 POLE, 2 MSI, 2 CNL, and 8 CNH subtype tumors). We also

presented the tree results based on SNVs/indels identified using exome sequencing, as well as those based on CNVs identified using target panel or exome sequencing (Supplementary Fig. 9; see also Supplementary Data 5 and 6). The phylogenetic trees of the CNL and CNH subtype tumors had a high proportion of trunks: the carcinoma and sarcoma components shared most of the SNVs/indels of the 596 genes (Fig. 5 and Supplementary Table 4; 15 of 16; 93.8%, and 56 of 74; 75.7% mutations in total). For these tumors, all driver events exclusively occurred on the trunk and many of the drivers were clonal (CCF ≥ 0.8) (Fig. 5 and Supplementary Table 4; 4 of 9; 44.4% and 23 of 27; 85.2% mutations in CNL and CNH subtypes). Yet, in the hypermutator (POLE and MSI) subtypes, the phylogenetic trees had shorter trunks and longer branches, with additional driver events on these branches. Whereas none of the 26 (0%) and only 2 of the 12 (16.7%) drivers on the carcinoma- or sarcoma-branches were clonal among the POLE and MSI subtypes, 21 of 43 (48.8%) and 9 of 12 (75.0%) driver mutations on the trunk were clonal, respectively (Fig. 5 and Supplementary Table 4; Fisher exact test $p < 0.0001$ and $p = 0.0123$). These results suggest less biological significance of the driver events on the branches for the POLE and MSI tumors. CS tumors of the CNH subtype frequently had longer trunks with shorter branches for both carcinoma and sarcoma components, and this branching pattern was also confirmed with tree analyses with SNVs/indels from the exome data (when available) and with CNVs of the target panel or exome data (Supplementary Fig. 9). No recurrent genetic mutations in SNVs/indels or CNVs were differentially detected between the carcinoma and sarcoma components in any of the tumors for which either the 596-gene panel or the exome data was examined (see Supplementary Note 3, Supplementary Table 4,

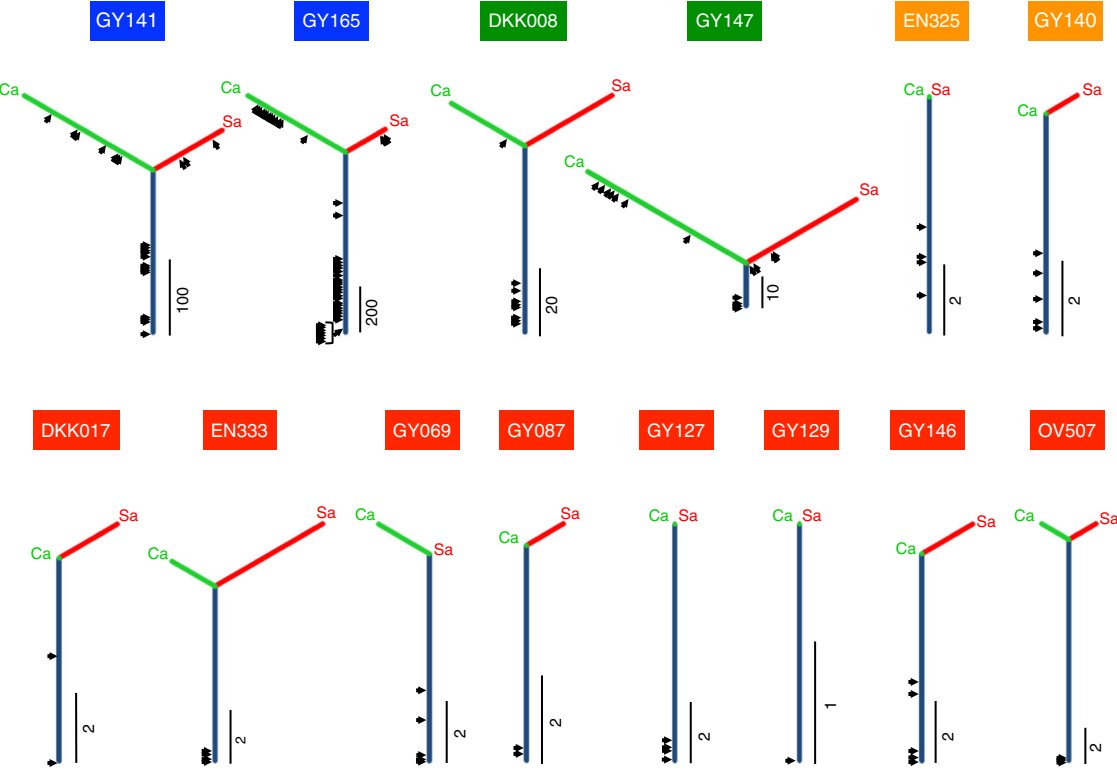

**Fig. 5** Clonal evolution in the carcinoma and sarcoma elements inferred from SNVs/indels based on the 596-gene panel. Phylogenetic trees are used to represent genetic similarity of the carcinoma and sarcoma components in the primary tumor. A trunk (navy) and a branch (yellow green; carcinoma and cherry color; sarcoma) represent shared and private mutations, respectively. The length of the trunk/branch is proportional to the number of SNVs and indels, and an index beside each tree indicates the number of base pairs. The timing of driver mutation acquisition is shown with an arrow. A case ID is shown on a tree with the color indicating the genomic aberration subtype: blue, POLE; green, MSI; red, CNH; and orange, CNL). Ca, carcinoma; Sa, sarcoma

and Supplementary Data 5 and 6). The clonal architecture analysis thus confirmed previous findings[13,14] that carcinoma and sarcoma components largely share critical driver mutations, and supports the combination or conversion—but not collision—theories for the histogenesis of CS.

**Genetic similarity of multiple regions in a tumor.** The extensive similarity in the mutational patterns of adjacent carcinoma and sarcoma elements (Fig. 5) may be caused by sampling proximity or by insufficient resolution due to panel sequencing with a limited number of genes. To further elucidate the clonal architecture of CS tumors, we conducted exome sequencing analyses of cancer cells collected from multiple regions with variable compositions of carcinoma and sarcoma elements for four tumors representing each of the four genomic aberration subtypes (Fig. 6 and Supplementary Table 5, also see the Methods). In the POLE subtype tumor (EN676), both the carcinoma-only T1 and carcinoma-dominant T2 branches were distant from the sarcoma-only T4 branch, which may indicate early bifurcation of both elements in the tumor. However, in the CNH subtype (GY030 tumor), similarities between T2 (carcinoma-dominant) and T3 (sarcoma-dominant), and between T4 (carcinoma-dominant) and T6 (sarcoma-dominant) suggest parallel evolution of the bimodal T2/T3 and T4/T6 regions. Similarly, in the CNL subtype (EN558 tumor), carcinoma-dominant T1 was closer than carcinoma-only T5/T7 in mutational distance to sarcoma-dominant T8/T2/T4/T3, which implies parallel evolution of the carcinoma components. Here, the genomic distances do not

correlate with carcinoma or sarcoma differentiation but simply reflect the physical distances between sampling positions within the tissue (Fig. 6). Clonal drivers consistently reside on the trunk, with few arising across the 23 private branches of the four tumors, similar to that outlined in Fig. 5. The only exception was T5 in the MSI subtype (EN482), in which two clonal driver mutations were detected (Supplementary Table 5). No recurrent genetic mutations in SNVs/indels or CNVs were differentially detected between carcinoma- and sarcoma-dominant regions with multiregional sequenced data of 3 cases (EN676, GY030, and EN558; the data from EN482 were not used for this purpose since sarcoma-dominant region sequenced data was not available for the case; also described in Supplementary Note 3). These observations again support the notion that carcinoma and sarcoma elements share driver event(s) in tumorigenesis. Thus, multiregional exome sequencing analyses revealed genomic alteration-independent differentiation of CS cells in tumors.

**Epithelial–mesenchymal transition in CS.** As described above, the carcinoma and sarcoma elements showed a high mutational similarity, particularly in terms of driver genes, and supported the hypothesis of the clonal origin of CS cells. Several studies have proposed EMT as a potential mechanism to convert carcinoma cells into sarcoma cells within a CS tumor[14,26,35]. We sought to know the impact of EMT on clinical variables and on several subtyping schemes, including genomic aberration-based molecular subtyping. For this purpose, we computed an EMT score from 81 genes for each CS sample with RNA-seq data according

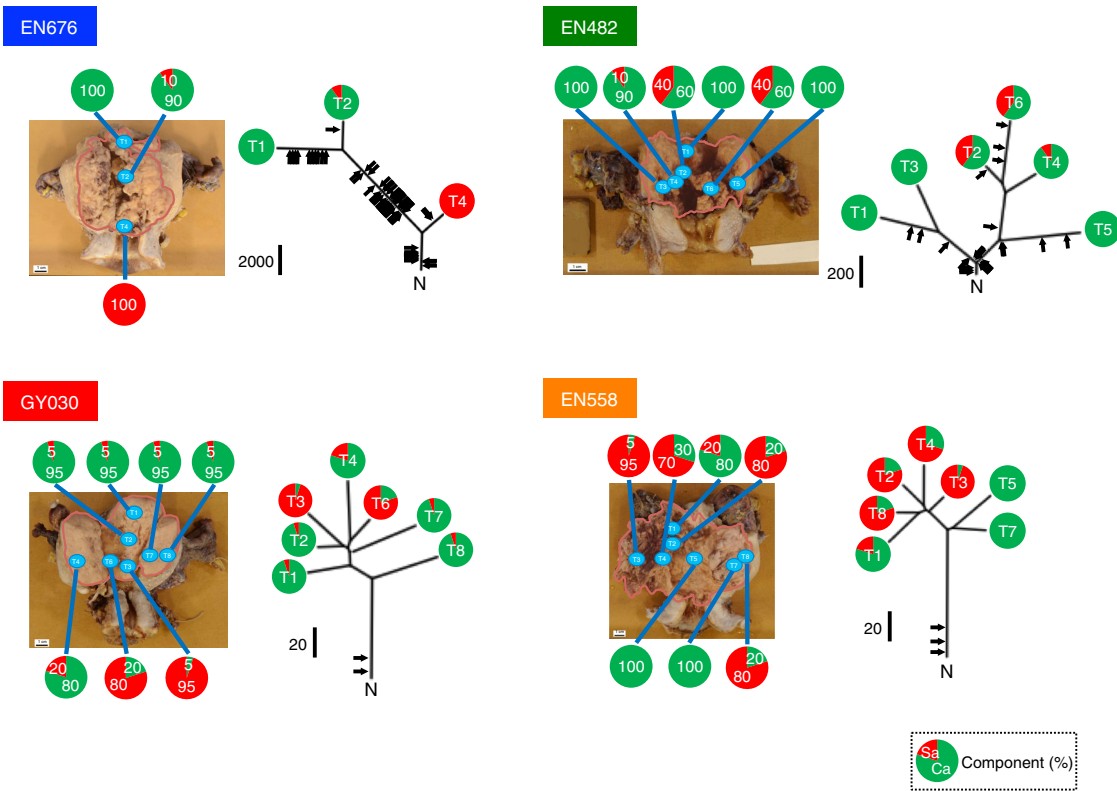

**Fig. 6** Phylogenetic tree analyses with SNVs and indels from the multi-regional exome-sequenced samples of uterine CSs. We selected representative tumors that were completely resected during surgery for each genomic aberration subtype (blue, POLE; green, MSI; red, CNH; and orange, CNL) and rendered to multi-regional exome sequencing. The phylogenetic tree (right) is presented alongside the entire tumor (left) with the margin shown as a pink line. The positions of sample collection are shown. Trunk/branch length is proportional to the total number of SNVs and indels. An index black bar beside the trunk indicates the number of base-pair mutations. The timing of driver mutation acquisition is shown with an arrow. Ca, carcinoma; Sa, sarcoma. The proportions of carcinoma and sarcoma components in each CS sample are shown as a pie chart, with green and red colors indicating the proportion of each component

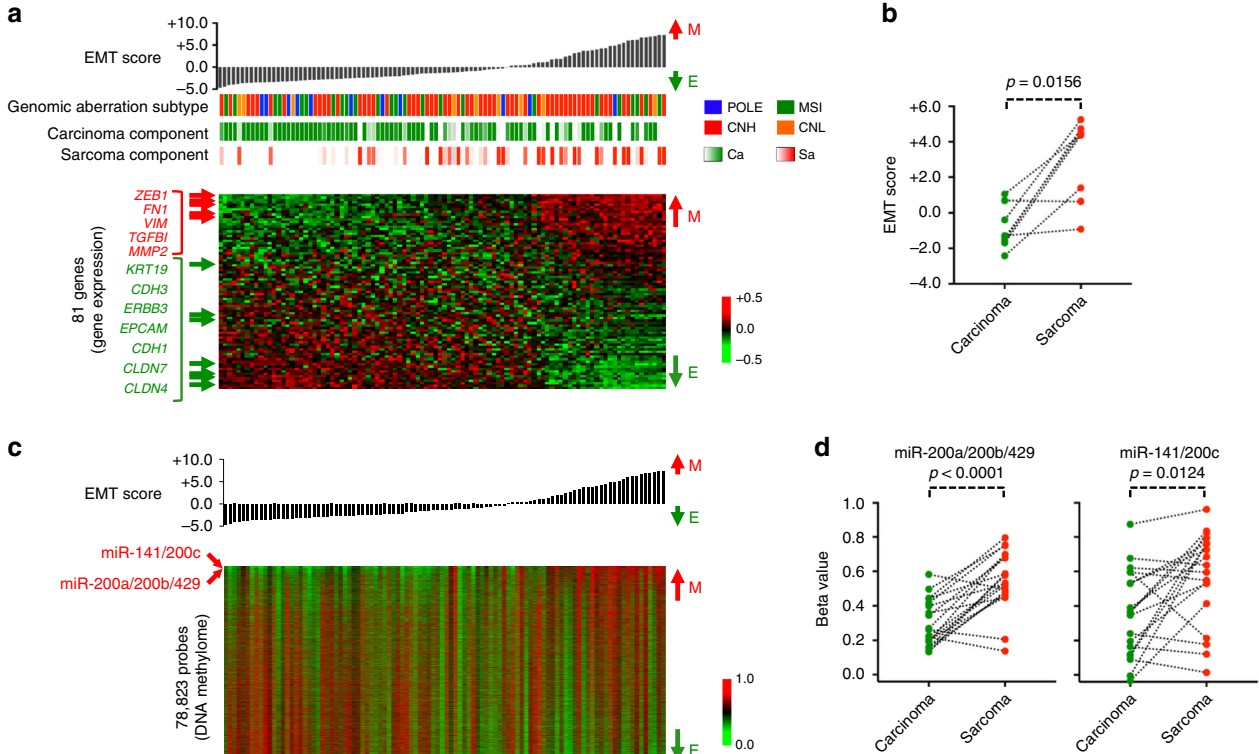

**Fig. 7** Epithelial-mesenchymal transition (EMT) in the transcriptome and DNA methylome of carcinosarcoma (CS) tumors. **a** Relationship among EMT score, EMT gene expression, genomic aberration subtype, and carcinoma or sarcoma content in a CS tumor. CS samples are sorted according to EMT score, calculated by the first principal component of 81 EMT marker genes. Red and green indicate high and low expression of the genes in the heatmap. Representative epithelial and mesenchymal genes are shown to the left of the heatmap in green and red, respectively. Color codes for the genomic aberration subtype are as follows: POLE, blue; MSI, green; CNH, red; and CNL, orange. Carcinoma and sarcoma content in a CS sample is shown by gradients of green and red, respectively. E; epithelial, M; mesenchymal. **b** EMT score of micro-dissected carcinoma or sarcoma components. The EMT scores were computed with RNA-seq data derived from the carcinoma and sarcoma components separately captured by laser capture microdissection. The Wilcoxon signed rank test was used to evaluate statistical significance. **c** DNA methylome correlated with EMT score across CS samples. CpG probes were selected by variable methylation within the top 20% variance. The resultant 78,823-probe $\beta$-values were sorted according to EMT score and are shown as a heatmap. The positions of CpG sites for miR-141/200c and miR-200a/200b/429 are indicated beside the heatmap in red. **d** miR-200a/200b/429 and miR-141/200c expression of micro-dissected carcinoma or sarcoma components. The Wilcoxon signed rank test was used to evaluate statistical significance

to a previous method[36]. The EMT score was highly concordant with the content of the carcinoma or sarcoma elements in the primary bulk tumor (Fig. 7a; Spearman $rho = -0.3140$, $p = 0.0016$). When the carcinoma and sarcoma components were separated, we found a significant distinction in EMT score (Fig. 7b). The EMT score was not significantly correlated with any clinicopathological parameter, such as histological grade and FIGO stage ($p = 0.5237$ and 0.8045 by Mann–Whitney U-test), or relapse-free survival and overall survival ($p = 0.4890$ and 0.3724 by Cox regression analyses) in the current cohort. Furthermore, genomic aberration subtype was not correlated with EMT score. However, the transcriptome subtypes of TS3 and TS5, and the ES4 epigenetic subtype had significantly higher EMT scores than the other subtypes (Supplementary Fig. 10). A correlative analysis of the EMT score with the DNA methylation microarray data identified strong positive correlations with the CpG probes for miR-141/200c and miR-200a/200b/429, which have been suggested to play critical roles in the suppression of EMT by antagonizing EMT-inducing transcription factors, such as ZEB1 and ZEB2[26] (Supplementary Fig. 11). This finding supports the potential role of miR-141/200a/200b/200c/429 in the conversion of the carcinoma cells to sarcoma cells in CS tumors (see also Supplementary Note 4).

## Discussion

The genomic analysis of 373 uterine corpus endometrial adenocarcinomas (TCGA UCEC) by The Cancer Genome Atlas uncovered four genomic aberration subtypes: POLE, MSI, CNL, and CNH[16]. Such genomic subtypes are not found in some of the cancer types exemplified by high-grade serous ovarian carcinoma, even when using more than 300 samples[19]. This may be because there is less heterogeneity in the genome of serous carcinomas than endometrioid carcinomas[16,37]. One may anticipate observing genomic aberration subtypes within CS, which are equivalent to those identified within carcinoma counterparts. Nevertheless, previous research has failed to detect clear CS genomic heterogeneity, possibly due to small sample sizes or insufficient inter-tumor complexity within the cohort[14,24,26,29]. Here, by taking advantage of a large genomic cohort of 109 CS samples, four genomic aberration subtypes could be identified, which are mostly equivalent to those described in the analysis of data from TCGA UCEC. Whereas the proportions of POLE and MSI subtypes are similar to those in TCGA UCEC, most of the uterine CS samples in the current cohort are CNH instead of CNL. Given that the CNH subtype exhibited an aggressive nature and was associated with unfavorable outcomes in endometrial carcinoma[16], this predominance of CNH in the CS cohort may

explain the aggressive behavior and poorer prognosis of CS[7]. It is also noteworthy that most of the carcinoma components of TCGA UCS samples[26] were serous and undifferentiated (total 81.3%), whereas 85.0% of our uterine CS samples had endometrioid histology. Such histological bias, which can be derived from ethnic differences, possibly led to the differential identification of genomic aberration subtypes within our cohort. Importantly, the same classification scheme could also detect all four of the subtypes in TCGA UCS samples with different proportions; albeit, such subtypes were not described in the initial study[26]. Given that MSI and POLE subtypes have been included in previous genomic work[13,22,24,25] and that the composition of the MSI subtype in the current cohort (22.0%; 24/109) is comparable with that in previous studies (5–40%)[11–13,20–25], we believe that this genomic subtyping scheme for CS is robust and reproducible.

In the current study, multiple differences in clinicopathological features such as FIGO stage, histological grade and patient outcomes were detected between ovarian and uterine CSs as previously described[4], which may be linked with the differential transcriptomic nature. Whereas OCS comprises mostly CNH tumors, UCS is more heterogenous in genomic aberration subtypes. This relationship mirrors that between serous ovarian carcinoma and endometrial endometrioid carcinoma, which exhibit less and more heterogeneity in genomic aberrations across samples. Such resemblance of OCS to serous ovarian carcinoma and UCS to endometrial endometrioid carcinoma may be derived from sharing cell of origin in each anatomical site.

There has been remarkable progress recently in drug development, and this has helped to identify the clinical relevance of DNA repair deficiencies in cancer; for instance, tumors with HRD show high sensitivity to PARP inhibitors through synthetic lethality, and immune checkpoint inhibitors are more effective on tumors with the hypermutator phenotype, such as microsatellite instability by MMR deficiency. It would therefore be reasonable to stratify patients with CS by genomic aberration subtype for the delivery of an individualized regimen as part of a model of precision care. Indeed, in a case report[38], olaparib, a PARP inhibitor, exhibited significant efficacy in a patient with ovarian CS harboring a germline mutation in *RAD51D*, a gene involved in HR repair, supporting the use of PARP inhibitors in patients with CS with HRD. Since driver mutations appear to be shared between the carcinoma and sarcoma elements, we anticipate that a molecularly targeted therapy could show equivalent efficacies against both components when an actionable target exists in the tumor.

Multiple driver gene mutations for SNVs, indels, and CNVs in CS have been identified in previous and current genomic studies;[13,14,22,24,26,29] the current study, however, relies mostly on targeted panel sequencing, which limits the identification of potential novel drivers. Consistent with the biological similarity between CS and the carcinoma counterpart, most of the driver genes showed extensive overlap with those identified from TCGA UCEC study[16]. Among the drivers identified, *ARID1A* is of particular interest. *ARID1A* mutation was correlated with better patient outcomes when all cases were included in the analysis, because it was more frequently mutated in POLE and MSI subtypes, which show an intrinsically more favorable prognosis. On the contrary, in the CNH subgroup, patients with *ARID1A* mutant tumors showed substantially poor prognosis. Since *ARID1A* has been reported to be involved in DNA double-strand break repair[39], there is a possible interaction between DNA copy number aberration and ARID1A protein loss in CS aggressiveness.

Previous studies have employed various molecular markers to show the monoclonal origin of the carcinoma and sarcoma components in CS development. However, there remains contention as to which histogenic hypothesis—divergent evolution or metaplastic differentiation—is central. The selective sequencing of carcinoma and sarcoma components have confirmed a clonal origin for all genomic aberration subtypes, as noted in previous literature[13,14]. Exome sequencing of multiple regions with varied compositions of carcinoma and sarcoma components has revealed differentiation-unrelated clonal similarity: this argues against divergent evolution by genomic changes as the major histogenic process of this biphasic tumor but supports the theory of metaplastic conversion of CS cells from carcinoma to sarcoma and vice versa. Whereas no genetic event was detected in determining the carcinoma or sarcoma elements as described above, substantial differences were indeed observed in the transcriptome and DNA methylome, in particular, in the form of an EMT. The activation of the transcriptional program coincided with the hypermethylation of miR-200a/200b/429 and/or miR-141/200c promoters, previously highlighted as important in EMT[26,35], supporting a possible definitive role of the methylation status of these microRNAs in CS-cell sarcoma differentiation. Whereas *CTNNB1*-activating mutations have long been recognized as a driving mechanism of EMT in multiple cancer types, including uterine endometrioid carcinoma[40], in the current CS cohort and contrary to our expectations, the mutational status was independent of the EMT score but, instead, associated with hypomethylation of the miR-200a/200b/429 and miR-141/200c promotors. These observations indicate a complex regulatory mechanism involving DNA methylation on the microRNA promoters in CS cells, which should be the subject of future study.

## Methods

**Ethical approval.** Carcinosarcoma patients underwent surgery at the Cancer Institute Hospital, Kyoto University Hospital, or the Saitama Medical University International Medical Center between 1998 and 2015. Ethical approval was obtained from the internal review boards of the Japanese Foundation for Cancer Research, Kyoto University Hospital, and Saitama Medical University. Recruited patients provided written informed consent.

**Histopathological diagnosis.** The study cohort initially consisted of 169 (142 uterine and 27 ovarian) samples (Supplementary Note 1). Of these, 138 samples (114 uterine and 24 ovarian CSs) underwent further processing after being diagnosed as CS according to the World Health Organization (WHO) 2014 classification[6] by at least two independent pathologists with expertize in gynecological pathology. Staging was performed according to the 2008 modified International Federation of Gynecology and Obstetrics (FIGO) system[41]. The clinicopathological characteristics of the patients and samples, and the genomic and epigenetic analytical results are presented in Supplementary Fig. 1 and Supplementary Data 1 and 2. In the current cohort, 2 cases (OV594 and OV343) had synchronous endometrial and ovarian carcinomas (SEOC) and were associated with endometriosis in the ovary. For both cases, the ovarian tumors were histologically diagnosed as CS (Supplementary Table 1).

**Processing tumor samples.** Surgical specimens were subdivided into multiple pieces and processed for histopathological and immuno-histochemical examinations (fixed in formalin), or DNA sequencing, RNA expression and DNA methylation analyses (snap frozen). As described below, formalin-fixed paraffin-embedded (FFPE) samples were also used for target panel and exome analyses when separate or multi-regional sampling was performed. Fresh-frozen (FF) or FFPE cancer tissues were cut into 10-μm-thick sections, and cancer cells were selectively enriched by manual macrodissection or laser-capture microdissection with an LMD7000 (Leica), following the manufacturer's protocol.

For primary bulk tumor genomic, epigenetic, and transcriptomic analyses, DNA/RNA samples were extracted from FF tumor specimens (Supplementary Data 2).

Laser-capture microdissection was used to dissect and separate carcinoma and sarcoma components from the primary tumors of 14 cases (2 POLE, 2 MSI, 8 CNH, and 2 CNL). The details of the samples and assays are described in Supplementary Table 2. Briefly, 5 tumor samples were FF and the remaining 9 tumors were FFPE. Eight pairs of carcinoma and sarcoma were rendered to targeted resequencing, whereas exome analyses were performed for 6 tumors (Supplementary Table 2).

For 4 cases (1 POLE, 1 MSI, 1 CNH, and 1 CNL), spatial multi-sampling was performed in regions containing mixed carcinoma and sarcoma components in different proportions. A total of 23 areas in these 4 cases were macrodissected from

primary tumors (all FFPE) and subjected to exome sequencing (Supplementary Table 3). In all regions, the macrodissections were performed by a pathologist so that the tumor purity was greater than 60% in the area.

The proportion of carcinoma and sarcoma elements in each region (Figs. 6 and 7a) was determined by a pathologist based on the number of tumor nuclei; this analysis was performed for primary bulk tumors and for multi-regional samples.

**Matched normal samples.** Matched normal DNA samples were used in all germline and somatic variant calling for both targeted panel and exome sequencing. The sources of normal DNA per case (whole blood, buffy coat, tumor-free normal adjacent tissue or tumor-free lymph node, preservation method [FF or FFPE]) are described in Supplementary Data 3.

**DNA/RNA preparation for genomic and epigenetic analysis.** Total DNA from each tumor sample and matched normal tissue or whole blood was extracted using the QIAamp DNA Micro Kit (Qiagen), according to the manufacturer's instructions. DNA quality and quantity were checked using a NanoDrop 2000 (ThermoFisher) and Qubit 2.0 fluorometer (ThermoFisher). DNA samples ($n = 117$) that passed specific criteria (DNA purity [optical density 260/280 nm > 1.8], ratio of dsDNA/ssDNA concentration [>0.35] and dsDNA concentration [>50 ng/µl]) were further processed for targeted re-sequencing (or exome) and DNA methylation microarray analyses. Total RNA from each tumor sample was extracted using an RNeasy Micro kit (Qiagen), according to the manufacturer's instructions. RNA quality and quantity were checked with the NanoDrop 2000 (ThermoFisher) and Agilent 2100 Bioanalyzer (Agilent). RNA samples ($n = 101$) that passed the criteria for RNA purity ([optical density 260/280 nm > 1.3], RNA integrity number [>4] and DV200 [>70]) were further processed for RNA sequencing (RNA-seq) analyses. Details of the samples and assays are described in Supplementary Data 2.

**Design of targeted gene panels.** We designed custom panels, CS-A (491 genes; 2.605 Mbps) and CS-B (114 genes; 422.9 kbps) (Supplementary Data 4), using SureDesign (Agilent Technologies) with a 9-gene overlap between the panels (596 genes after deduplication). The genes were curated from previous literature, and included genes known to be mutated in endometrial and ovarian cancers and in sarcomas (rhabdomyosarcoma, liposarcoma, uterine leiomyosarcoma, and endometrial stromal sarcoma)[16,19,22,37]. We further added the actionable pan-cancer genes curated from commercially available targeted resequencing panels (FoundationOne [Foundation Medicine], Cancer Research Panel [Agilent] and Cancer Panel [Illumina]) and genes involved in DNA repair collected from databases[42] (Supplementary Data 4).

**Library preparation and sequencing analysis.** For primary bulk tumor analysis, a total of 109 sample pairs (tumor and matched normal specimens) were subjected to sequencing analysis (104 and 5 cases for CS panels and exome, respectively). DNA (200 and 1000 ng, respectively) was used for targeted re-sequencing and whole-exon sequencing. Following shearing, end repair, phosphorylation, and ligation of the barcoded adapters, DNA was subjected to hybrid capture with SureSelectXT Custom or SureSelect Human All Exon V4 or V5 kits (Agilent Technologies; the V4 and V5 kits were used before and after January 2013, respectively; Supplementary Data 2 and Supplementary Tables 2 and 3). The captured DNA was multiplexed and sequenced using an Illumina HiSeq2500 with a median coverage of 282–810 reads per tumor and 166–683 reads per normal specimen on the CS-A panel; 310–1936 reads per tumor and 261–1126 reads per normal specimen on the CS-B panel; and 225–354 reads per tumor and 80–197 reads per normal specimen for whole exons. The KAPA HyperPlus Kit (KAPA Biosystems) was used to construct libraries from DNA samples extracted from the FFPE tissues. When a combinatorial analysis of exome and target panel sequencing data was needed, exome data was down-sized with the merged BED file for CS-A and CS-B panels.

**RNA sequencing.** A total of 200 ng of RNA was converted into mRNA libraries using the TruSeq Stranded mRNA HT Sample Prep Kit (Illumina), following the manufacturer's instructions. Libraries were sequenced as $101 + 8 + 8 + 101$ bp with a dual-indexed run on an Illumina HiSeq2500 with a total of 33.7–131.4 million paired-end reads per tumor. RNA reads were aligned to the GRCh37 reference genome. Data with %Total Aligned >0.88 and %Abundant <0.3 were processed for further analyses ($n = 97$). Fragments per kilobase of exon per million reads mapped (FPKMs) as gene expression values were computed by RSEM-1.3.0[43], mapped by bowtie2-2.2.3 and annotated by ENSEMBL release 75. For further calculations, we transformed the FPKMs by $\log_{10}$ after adding a pseudo value of 1 to avoid an infinite value.

**Methylome analysis.** DNA methylation status was analyzed using 500 ng DNA from each tumor ($n = 108$) on the Infinium HumanMethylation 450 BeadChip Arrays (Illumina), according to the manufacturer's instructions. Highly variably methylated probes (the top 2,803 probes showing the top 2.5% in variance) on CpG islands of promoter regions were subjected to consensus clustering with unsupervised hierarchical clustering. Hypermethylated samples in the CpG islands of differentially methylated regions of *BRCA1*, *RAD51C,* and *MLH1* genes were

determined using probes selected as follows: (1) the maximum minus minimum values across the samples >0.4; and (2) the median value across the samples <0.1. An averaged $\beta$ value of 0.4 per sample was used as the cut-off.

**SNP6 analysis.** For 105 samples, somatic DNA copy number variants (CNV) in the tumor were detected with Genome-Wide Human SNP6.0 arrays, according to the manufacturer's instructions (Affymetrix). This was done in addition to informatics analysis of sequenced data with EXCAVATOR (described below). Bioconductor package rCGH (ver. 1.12.0) was used to compute the mean segmented copy number data. The circular binary segmentation (CBS) algorithm was used for segmentation, which is implemented in rCGH (ver. 1.12.0).

**Bioinformatics quality control for sequencing analyses.** Among the 117 CS samples that underwent targeted re-sequencing or exome analyses, 8 samples (7 uterine and 1 ovarian tumors) were omitted from further informatics processing because of poor tumor cellularity (less than 0.30 in tumor purity; described below). This change in sample number is described in a REMARK diagram (Supplementary Note 1 and Supplementary Fig. 2).

**Tumor purity.** Tumor purity was estimated using B-allele frequency in a loss of heterozygosity (LOH) region using ExomeCNV[44]. Tumor purity is shown as a color map in Fig. 1.

**Alignment and local realignment.** Sequenced reads were aligned (Burrows-Wheeler Aligner; ver. 0.6.1) to the human genome reference (hg19)[45]. GATK (GenomeAnalysisTK; ver 1.5-30) was used to recalibrate variant quality score and perform local realignment[46].

**Germline variant call and pathogenicity interpretation.** Germline SNV/indels were called with the GATK UnifiedGenotyper (ver 3.4.0)[47]. We considered germline variants with the following conditions as significant: 1) non-synonymous substitutions; 2) read depth >20; 3) read frequency >0.2; and 4) global minor allele frequency (MAF) <0.01 in dbSNP (http://www.ncbi.nlm.nih.gov/SNP/), ExAC (The Exome Aggregation Consortium; ver 0.3), HGVD (Human Genetic Variation Database; ver 1.42), or ToMMo (Tohoku Medical Megabank Organization; ver hum0015.v1). Pathogenicity of a germline variant was determined according to the American College of Medical Genetics and Genomics, Association for Molecular Pathology (ACMG-AMP) guidelines[48].

**Somatic variant call.** Somatic SNVs were called with VarScan (ver. 2.3.7), MuTect (ver.1.1.4), and Karkinos (ver. 3.0.22)[49–51], whereas somatic indels were detected with VarScan (ver. 2.3.7), SomaticIndelDetector (ver.1.5-30), and Karkinos (ver.3.0.22)[46,49,51]. SNVs and indels were taken as genuine mutations when they were detected by at least 2 of the 3 callers. For target panel analysis, somatic SNV/indels were first called with a pair of normal and tumor DNA samples using separate BAM files of CS-A and CS-B panels. Subsequently, the detected somatic variants on each panel were merged after deduplication of 9-gene variants.

Loss of heterozygosity (LOH) in copy number was determined by the number of variant sequence reads compared with the wildtype in the tumor DNA. Tumor samples with a loss of the wildtype allele were defined as those with a variant read frequency of >0.6[31].

Somatic CNVs were detected using EXCAVATOR (ver. 2.2) for sequencing results[52]. For target panel analysis, the BAM files of CS-A and CS-B panels were first merged for normal and tumor DNA samples separately. The merged BAMs were then used to detect somatic copy number aberrations in a tumor by comparing normal and tumor DNAs. The heterogeneous shifting level model (HSLM) algorithm implemented in EXCAVATOR was used for segmentation. The copy number status of each gene on the panel, as estimated with high-level thresholds in the GISTIC analysis, was highly concordant between SNP6 array outputs and EXCAVATOR results derived from targeted resequencing data; the percentage of CNV call concordance was 98.04% (median; range, 69.82 to 100%) for 105 samples. In the SNP6–GISTIC analysis, copy number (CN) changes were defined as amplification (CN ≥ 6), gain (CN = 3, 4, or 5), loss (CN = 1), or homozygous deletion (CN = 0), as per previous literature[53].

**Driver event exploration in a tumor.** Driver SNVs and indels were identified by statistical analyses with IntOGen software (IntOGen pipeline ver. 3.0.5, Inkscape 0.92)[33,54]. GISTIC (ver. 2.0.22) was used to define driver copy number segments[34]. As cut-off values, we used $q$-values less than 0.1, 0.2, and 0.1 for Oncodrive-fm, OncodriveCLUST, and GISTIC, respectively. A modified version of basicOncoPrint.R (gist.github.com/armish/564a65ab874a770e2c26) was used to present various types of genetic and epigenetic alterations in the CS samples.

**Tumor ploidy.** ABSOLUTE ver. 1.0.6[55] and FACETS ver. 0.5.14[56] were used to compute tumor ploidy for SNP6 array data ($n = 105$) and target panel or exome resequencing data ($n = 4$), respectively.

**Filters for FFPE noise**. Sequencing noise, mainly caused by damaged DNA in the FFPE samples, was filtered by: (1) removing any mutant alleles called at poorly mapped reads (mapping quality < 30); (2) removing any mutant alleles with a read depth <50; (3) removing any indels called at the edge of homopolymeric nucleotides (more than 4 of the same successive nucleotides); and (4) removing any recurrent mutant alleles across cases in the cohort but not recurrent in the TumorPortal or COSMIC database. Any mutant allele on aneuploid chromosomes was removed to accurately evaluate allele frequency. The same filters were applied to sequencing data from FF samples in the series. Reads of each hyperplasia or carcinoma sample from the same patient in the time-course series were subsequently subjected to pairwise local realignment using GATK in a pair with a matched normal sample to reduce erroneous calls by misalignment[57].

**Phylogenetic tree analysis**. Phylogenetic trees based on SNVs/indels were constructed with PHYLIP ver. 3.695 package[58]. To obtain the most parsimonious tree, we ran dnapars with the default parameters on somatic SNV/indels of the carcinoma and sarcoma components from the same patient[59]. The constructed trees were then drawn using Inkspace ver. 0.92, where the length of a trunk and a branch are proportional to the number of SNVs/indels. The timing of a driver event is inferred from the cancer cell fraction calculated from the mutant allele frequency of a SNV/indel and the copy number state where the SNV/indel resides[60]. A mutation was defined as clonal if the cancer cell fraction of the SNV/indel was more than 0.8.

Phylogenetic trees based on CNVs were constructed as follows: First, the Log2R of each probe per sample was computed using target panel or exome-sequenced data with EXCAVATOR ver. 2.2[52]. Second, multi-sample segmentation was performed with multipcf ver. 1.24.0 in R copynumber package[61]. Finally, TuMult[62] was used to construct a phylogenetic tree based on the segments, with integer copy number annotated with EXCAVATOR.

**Molecular subtyping based on genomic aberration profiles**. Following the classification scheme developed by TCGA for endometrial cancer (based on genomic aberration profiles), we made a decision tree to categorize the 109 CS samples (104 target-panel and 5 exome-sequenced data) into 4 genomic aberration subtypes (Supplementary Fig. 3). Exome data were down-sized with the merged BED file for CS-A and CS-B panels and used for subsequent combined analysis with target panel data. Samples were first categorized by *POLE* hotspot mutations in the exonuclease domain (*POLE* mutated; POLE) and then by MSI-high status (microsatellite instability; MSI), which was determined by deviations from paired normal control in electropherograms of 2 or more among 6 DNA markers (BAT25, BAT26, D2S123, D5S346, D17S250, and BAT40), as previously described[63]. We determined copy number high (CNH) subtype based on DNA copy number aberrations using target panel or exome sequencing data. For this, we first selected the regions that were used for molecular subtyping in the original TCGA analysis[16] from the whole target panel region. We then performed unsupervised consensus clustering of the mean segmented copy number data. We found that CNH tumors congregated as a cluster that resembled the cluster 4 in TCGA analysis[16]. Copy number low (CNL) was finally assigned to the remaining samples.

**Molecular subtyping based on driver mutation profiles**. We used another molecular subtyping method based on major driver mutations observed in carcinomas with endometrioid or serous histology[22,26]. For this analysis, samples with *PTEN* or *ARID1A* mutations were assigned as endometrioid-like, samples with *TP53* or *PPP2R1A* mutations were classified as serous-like, and the remaining samples lacking any mutation in these four genes were assigned to the unclassified group.

**TCGA data analyses**. We downloaded copy number data (masked copy number segment data from Affymetrix SNP 6.0), expression data (RNA-seq data normalized by Fragments Per Kilobase of exon per Million mapped fragments [FPKM]), and DNA methylome data (methylation beta-value data from Illumina Human Methylation 450 or Human Methylation 27) for 373 uterine corpus endometrial cancer (UCEC), 57 uterine carcinosarcoma (UCS) and 273 high-grade serous ovarian cancer (OV) samples generated by TCGA from Genomic Data Commons Data Portal (https://portal.gdc.cancer.gov) for the analysis described in Fig. 4. For the expression analysis, the FPKMs were transformed by log10 after adding a pseudo value of 1 to avoid an infinite value. Information of somatic SNV/indels per sample in TCGA UCEC, TCGA UCS, and TCGA OV was obtained through cBioPortal (http://cbioportal.org) and used for further analyses including genomic aberration and driver mutation subtyping as described above.

**EMT score and micro-environmental analysis**. Following the previous analysis in TCGA UCS study[26], the first principal component of the 81 EMT marker gene expression data was used as the EMT score[36]. In the current cohort, the EMT score exhibited strong correlations with the other two different EMT scores developed by Mak et al.[64] and by Tan et al.[65] ($Rho$ = 0.7999 and 0.9360; $p$ < 0.0001 and $p$ < 0.0001 by Spearman correlations of 97 CS RNA-seq data; the first principal component of each signature was used for simplicity), indicating robustness of the score.

**Statistical analyses**. Consensus Clustering[66] was employed to identify clusters corresponding to distinct subgroups in the CS transcriptome or DNA methylome data using R (http://www.R-project.org) with Bioconductor ConsensusClusterPlus[67]. We chose k-means clustering algorithm with Euclidean distance and a sub-sampling ratio of 0.8 for 1000 iterations. Mann–Whitney *U*-test and Fisher's exact test were used to statistically evaluate correlations of clinicopathological parameters with genomic subtypes using GraphPad Prism or R software. The log-rank test was employed in Kaplan–Meier analyses for overall survival and relapse-free survival with GraphPad Prism. Univariate or multivariate Cox proportional hazards regression models (R software) were used to identify prognostically relevant mutations, expression, or various biological scores. Before assessing the prognostic association, the expression and biological score data were transformed to binary information; the presence and absence of an alteration were determined based on values above and below a cut-off value, respectively, when the information was given as a numeric.

**Reporting summary**. Further information on research design is available in the Nature Research Reporting Summary linked to this article.

## Data availability
Exome/target panel sequencing data have been deposited in the National Bioscience Database Center (NBDC; https://biosciencedbc.jp/en/) under the accession number JGAS00000000172. RNA sequencing data have been deposited in the NCBI Gene Expression Omnibus (GEO; https://www.ncbi.nlm.nih.gov/geo/) under the accession number GSE128630. The TCGA uterine CS dataset was accessed from the Genomic Data Commons [https://portal.gdc.cancer.gov/projects/TCGA-UCS]. DNA methylation data have been deposited in the NCBI Gene Expression Omnibus (GEO; https://www.ncbi.nlm.nih.gov/geo/) under the accession number GSE136790. All the other data supporting the findings of this study are available within the article and its supplementary information files and from the corresponding author upon reasonable request. A reporting summary for this article is available as a Supplementary Information file.

## Code availability
In the current study, we extensively used publicly available algorithms and did not use any custom code.

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

## Acknowledgements

We thank Kazuma Kiyotani, Tsukasa Baba, Yasuo Uemura, Yu Imamura, Takako Yokomizo, and Futoshi Akiyama for helpful discussions. We also thank Yukie Naka-shima, Megumi Nakai, Rika Nishiko, Junko Kanayama, Nobuyuki Fukui, Akihisa Takahara, Tomoko Kaneyasu, Sayuri Amino, Yuki Ota, Noriko Yaguchi, and Kumiko Sakurai for technical assistance; Minako Hoshida for administrative assistance; and Rebecca Jackson for editing a draft of this manuscript. This work was supported by JSPS

KAKENHI; Grant Numbers JP17K18337, JP15K06861, JP18K07338, JP26462543, and JP17K11308 and by the Vehicle Racing Commemorative Foundation; Grant Numbers 5144, 5274, and 5393.

## Author contributions

O.G., Y.S., and S.M. performed the analyses and wrote the paper. Y.T. confirmed the histopathological diagnoses. N.T. analyzed the data. Y.S., K.K., K.O., N.T., H.N., K.H., K. F., M.T., and N.M. collected the specimens and provided clinical information. T.N. and S. M. conceived the study and wrote the paper.

## Competing interests

The authors declare no competing interests.
