## [Peer Review File · Nature Communications]

Reviewers' comments:

Reviewer #1 (Remarks to the Author): Expert in carcinosarcoma

This manuscript describes the characterisation of a large cohort of uterine and ovarian carcinosarcoma samples collected at two Japanese institutions over a 17 year period (1998 – 2015). Samples were reviewed by two gynaecological pathologists.

Analysis consists of panel (105 samples; 596 genes, median read depth >250x) or exome sequencing (4 samples; >225x) on tumour:normal pairs, RNAseq (100 samples; number of reads per sample not stated) and DNA methylation (108 samples; Infinium 450 array). There was no dedicated assessment of copy number using SNP6 array or WGS methods, although CNV were called from the panel sequencing data. Bioinformatically, standard scripts and packages were utilised, and classification was primarily based on the TCGA uterine results.

Several major publications have previously identified the genomic, copy number, transcriptomic and methylation profiles of uterine carcinosarcoma (together with a smaller number of ovarian carcinosarcomas), in particular Zhao et al (ref 23), Jones et al (ref 37) and Cherniak et al (ref 39). Thus, this study struggles for originality/novelty – I acknowledge that this study has a larger sample size than previous publications, but there is no greatly original new analysis.

Overall, the work has been performed to a high standard and the data are presented clearly. As stated above, the results are largely confirmatory; there are few mutations that differ between the carcinoma and sarcoma elements of each tumour and that there is very late clonal divergence between carcinoma and sarcoma; the uterine samples in particular can be mapped onto the TCGA endometrial carcinoma classifiers; defective homologous recombination may be associated with improved outcome; EMT is a definite feature with evidence of aberrant methylation around specific miR clusters, including miR-141/200c and miR-200a/200b/429.

One unusual feature is the high rate of endometrioid histology in both the endometrial and ovarian groups – far higher than in previous publications. The samples were reviewed by gynaecological pathologists and met the WHO 2014 criteria for carcinosarcoma; and the authors suggest that there might be ethnic drivers of the different histotype mixture.

Comments

1. The inclusion of the ovarian cohort and the use of an endometrial cancer classifier. The authors state that it is possible to fit these samples into the 4 clusters identified by the TCGA UCEC analysis. However, this reviewer would argue against including the ovarian samples in the same analysis – for example the prognosis of the ovarian cohort was worse than the endometrial cohort (Supplementary Figure 1B, although the proportion of stage I was much higher in the endometrial cases compared to the ovarian). Ovarian/tubal high grade serous carcinoma and CNH/serous endometrial cancer have many similarities, but they are not the same. The appropriate comparators for the ovarian samples are the large ovarian cancer datasets, including TCGA, PCAWG/IGCS etc.
2. Copy number. Given that it is clear that ovarian/tubal high grade serous carcinoma and CNH/serous endometrial cancers both have highly complex CN landscapes, detailed CN analysis (beyond that which can be called from panel sequencing) is required. Previous analyses have noted the very high rates of whole genome duplication in uterine carcinosarcoma for example.
3. A REMARK diagram is essential: 169 samples were originally identified, of which 138 passed pathology review. 117 passed DNA quality control and genomic sequence data from 109 are presented. Only 100 underwent RNAseq. These data are buried in the supplementary methods and should be readily visible to the reader with an explanation for why samples were discarded at various stages of the process.
4. I am curious that a POLE-mutated ovarian sample was identified. The rate of POLE mutations in ovarian cancer is very low (2 validated in the TCGA cohort for example) and I wonder whether this

sample might have been a misclassified endometrial primary..

5. Iniparib is no longer considered to be a PARP inhibitor, so the relevant comment (end of page 11, start of page 12) should be removed. Moreover, as a result, the negative trial of iniparib (ref 64) in uterine CS proves nothing about whether or not defective HR is a meaningful therapeutic target in CS. This whole sentence should be removed.

Reviewer #2 (Remarks to the Author): Expert in gynaecological cancer genomics

This descriptive paper presents data from an exhaustive genomic analysis of CS with the aim of resolving its origin and key driver genes. The paper succeeds in convincingly clarifying that CS are not collision tumours although this had partly been shown in previous studies. The other significant finding is that CS cancer fall into 4 molecular subtypes that are reflective of those already described for endometrial cancers analysed by the TCGA. It is interesting that the sarcoma component of the tumours are associated with expression of genes known to be involved in EMT. Overall this paper provides some interesting insight into this sub-type of endometrial/ovarian cancers, although the clinical implications of this work remain unclear and not really addressed by the authors.

A limitation of the study is the use of only a targeted sequencing panel which has been useful in clonality analysis and understanding the role of known driver genes but does not represent a defining study of all driver genes. Related to this is the lack of clarity as to how the sequencing, expression and methylome data was generated. Maybe I missed it, but the only description I could find was the targeted and exome sequencing platform but nothing about the expression and methylation. In addition, how many of the cases were FFPE and fresh frozen. There is no mention of access to matching normal DNA so how were somatic mutations distinguished from germline variants. These details are required in order to assess the robustness of the actual data and subsequent interpretations.

Related to the point above, In figure 2, how were the somatic and germline mutations identified

The authors present data showing that the sarcoma components of the CS tumours show characteristic features of EMT, which is in fact quite expected as they are mesenchymal. But are the authors suggesting the methylation of MIR200 etc are driver events? I suspect not but this needs to be stated explicitly. It would seem logical that some other influence is driving EMT and the gene expression changes are simply reflecting that new induced state. The authors did do whole exome sequencing on purified epithelial and sarcoma components of 4 CS tumors but do not report if there were any sarcoma specific alterations that might explain the EMT. Even if none were found this should be stated.

The authors should state that CS were previously known as a malignant mixed mullerian tumor . This provides some historical context.

minor spelling/ grammatical errors.

Page 6 there is an "of" missing: alteration-independent OF carcinoma

Page 7. microsatellite instable (MSI) should be microsatellite instability.

Page 16. "These results suggest less biological significance of the driver... should be BIOLOGICAL

Page 17. each tumor samples represent one genomic . should be SAMPLE

Reviewer #3 (Remarks to the Author): Expert in cancer evolution

In this manuscript entitled "Clinically-relevant Molecular Subtypes and Genomic Alteration-independent Differentiation in Gynecologic Carcinosarcoma" by Gotoh et al., the authors performed multi-omics data analyses for a relatively large dataset of carcinosarcoma and clarified four molecular subtypes based on genomic alterations. They also demonstrated that epithelial mesenchymal transition (EMT) was a key mechanism of sarcoma differentiation by using transcriptome and DNA methylome analyses. In addition, comparison of genomic/transcriptomic alterations between their dataset and TCGA dataset is highly evaluated in terms of enhancement of this study's quality. However, they did not show the detail of this study in the manuscript including supplementary information. The reproducibility and reliability of the results are most important in bioinformatics field. In particular, they should describe "materials and methods" in much more detail. Detailed comments are described below.

1. Abstract (Page 2): Carcinosarcoma of the uterus is not so rare compared to of the ovary.
2. Abstract (Page 2, Lines 6-7): The authors described as "Panel sequencing of 109 CS with 596 genes" in the abstract. However, in Supplementary Materials and Methods, they described as "105 and 4 cases for CS panels and exome, respectively".
3. Abstract (Page 2, Line 8): The authors should avoid using non-recognized abbreviations such as "CNH and CNL".
4. Abstract (Page 2, Line 13): This sentence ("The current study thus provides novel therapeutic options for CS") is apparently overstated. The authors should conclude based on their results.
5. Introduction (Page 5, Lines 6-8): TCGA pan-cancer study demonstrates that highly enriched events are primarily mutations in uterine corpus endometrial cancer (PMID: 24071851).
6. Results (Page 7, Lines 3-4) and Supplementary Materials and Methods (Suppl. M&M) (Page 16): How many samples were used in the molecular subtyping? Did the authors use both targeted and exome sequencing data in this analysis? If so, how did you merge two types of data into one? Data quality of targeted genes are different between targeted sequencing data and exome sequencing data.
7. Results (Page 7) and Suppl. M&M (Page 16): How did the authors determine MSI-high status?
8. Results (Page 7) and Suppl. M&M (Page 14): Several methods (including EXCAVATOR) to detect copy number alterations from exome sequencing data have a limitation to infer copy number across whole genome because of no use of sequencing reads from intergenic or intronic regions. On the other hand, copy number alterations is one of the most important information in this study. Therefore, the authors should validate copy number alterations by other modalities.
9. Results (Page 9, Lines 4-5): Although the authors used data derived from 57 TCGA uterine CS, they did not describe the detail of TCGA data in the manuscript as well as Suppl. M&M. Which type of data did they use? Did they download FASTQ or BAM files from GDC and detect mutations and copy number alterations by themselves?
10. Results (Page 12, Lines 14-15): The authors should describe a proper reference.
11. Results (Page 13, Lines 3): Again, the authors should avoid using a non-recognized abbreviation "JFCR" before any explanation.
12. Results (Page 13): Again, the authors did not describe the detail of TCGA data in the manuscript as well as Suppl. M&M.
13. Results (Page 15) and Figure 5: The authors demonstrated phylogenetic trees by using 596-gene panels. However, CNH samples harbor little SNV/indels and estimation of clonal evolution based on SNV/indels is not suitable for CNH samples. They could perform estimation of clonal evolution based on copy number alterations if they succeeded to validate copy number alterations by other modalities. And they also could show similarities or differences in mutation and copy number alteration profiles between carcinoma and sarcoma components in the same tumor as a supplementary table. This information might be useful when they will discuss about therapeutic strategies for CS.
14. Results (Page 17) and Suppl. M&M (Page 9): How did the authors determine the percentages

of carcinoma and sarcoma components? How about tumor purity?

15. Materials and Methods (Page 24) and Suppl. M&M (Page 12): Why did the authors use two types of SureSelect (v4 or v5)? They should clarify which data was based on SureSelect v4 or v5.

16. Suppl. M&M (Page 9): The authors should describe tumor purity in each sample used in genomic analyses. Genomic data based on bulk sample is influenced by tumor purity.

17. Suppl. M&M (Page 10): The authors should describe which type of normal tissue was used. And they should also describe the information of matched normal sample per case. Selection of normal sample is very important for copy number detection.

18. Suppl. M&M (Page 12): It is well known that RPKM is not good to compare across samples. It may be better to select Transcripts Per Kilobase Million (TPM) compared to RPKM or FPKM.

Response to Comments

Reviewer #1 (Remarks to the Author)

This manuscript describes the characterisation of a large cohort of uterine and ovarian carcinosarcoma samples collected at two Japanese institutions over a 17-year period (1998–2015). Samples were reviewed by two gynaecological pathologists.

Reviewer #1 summarized our research structure for sampling in the current study. We would like to point out that the samples were collected at three Japanese institutions (the Cancer Institute Hospital, Kyoto University Hospital and Saitama Medical University International Medical Center). This description appears in the section, “Ethical approval” in the “Materials and Methods” in the Main document.

Analysis consists of panel (105 samples; 596 genes, median read depth >250x) or exome sequencing (4 samples; >225x) on tumour: normal pairs, RNAseq (100 samples; number of reads per sample not stated) and DNA methylation (108 samples; Infinium 450 array). There was no dedicated assessment of copy number using SNP6 array or WGS methods, although CNV were called from the panel sequencing data. Bioinformatically, standard scripts and packages were utilised, and classification was primarily based on the TCGA uterine results.

We obtained the RNA-seq data for a total of 33.7–131.4 million paired-end reads per tumor. Following the Reviewer’s suggestion, we include this description in the section titled “RNA Sequencing” in the “Supplemental Materials and Methods”.

This comment regarding our CNV analysis is related to Comment #2 raised by the same Reviewer as well as Comment #8 raised by Reviewer #3. Because of these comments, we realized that we needed to confirm the reliability of the detected CNVs from the target panel sequencing data by comparing it with those from the SNP6 arrays, which we additionally performed with 105 CS samples for which DNA remained available. We describe the results from the SNP6 arrays in the specific Comment #2 below made by this Reviewer and in response to Comment #8 from Reviewer #3.

Several major publications have previously identified the genomic, copy number, transcriptomic

and methylation profiles of uterine carcinosarcoma (together with a smaller number of ovarian carcinosarcomas), in particular Zhao *et al.* (ref 23), Jones *et al.* (ref 37) and Cherniak *et al.* (ref 39). Thus, this study struggles for originality/novelty – I acknowledge that this study has a larger sample size than previous publications, but there is no greatly original new analysis.

Overall, the work has been performed to a high standard and the data are presented clearly. As stated above, the results are largely confirmatory; there are few mutations that differ between the carcinoma and sarcoma elements of each tumour and that there is very late clonal divergence between carcinoma and sarcoma; the uterine samples in particular can be mapped onto the TCGA endometrial carcinoma classifiers; defective homologous recombination may be associated with improved outcome; EMT is a definite feature with evidence of aberrant methylation around specific miR clusters, including miR-141/200c and miR-200a/200b/429.

One unusual feature is the high rate of endometrioid histology in both the endometrial and ovarian groups – far higher than in previous publications. The samples were reviewed by gynaecological pathologists and met the WHO 2014 criteria for carcinosarcoma; and the authors suggest that there might be ethnic drivers of the different histotype mixture.

Comments

1. The inclusion of the ovarian cohort and the use of an endometrial cancer classifier. The authors state that it is possible to fit these samples into the 4 clusters identified by the TCGA UCEC analysis. However, this reviewer would argue against including the ovarian samples in the same analysis – for example the prognosis of the ovarian cohort was worse than the endometrial cohort (Supplementary Figure 1B, although the proportion of stage I was much higher in the endometrial cases compared to the ovarian). Ovarian/tubal high grade serous carcinoma and CNH/serous endometrial cancer have many similarities, but they are not the same. The appropriate comparators for the ovarian samples are the large ovarian cancer datasets, including TCGA, PCAWG/IGCS etc.

Following this suggestion, in Figure 4, we separated the ovarian CS omics heatmaps from those for uterus-originating CS and show the ovarian results beside those from TCGA high-grade serous ovarian cancer (TCGA OV) ($n = 273$) (Cancer Genome Atlas Research 2011). The distinctive nature of the ovarian and uterine CS samples became clearer after

this separation in the transcriptome. We have modified the description in the section, “Molecular features of genomic aberration subtype” in the Results accordingly. Also, to avoid seeing any possible confounding effect of the ovarian CS on prognosis, additional prognostic correlation analyses were performed with exclusion of the ovarian data and the results were described in sections of “Identification of four distinct genomic subtypes in CS samples” and “Genomic instability and DNA repair defects in CS subtypes” in the Results.

2. Copy number. Given that it is clear that ovarian/tubal high grade serous carcinoma and CNH/serous endometrial cancers both have highly complex CN landscapes, detailed CN analysis (beyond that which can be called from panel sequencing) is required. Previous analyses have noted the very high rates of whole genome duplication in uterine carcinosarcoma for example.

As mentioned above, we performed additional SNP6 arrays with 105 CS samples for which DNA remained available to detect somatic CNVs in the tumors.

In appreciation of the high resolution and wide coverage of the SNP6 data, we conducted two additional analyses based on the SNP6 CNV information to better characterize the CS tumors.

GISTIC analysis of the SNP6 copy number data changed the result of driver CNV calling from that determined with the panel sequencing data. We made an Oncoprint figure for this result with a smaller sample size ($n = 105$) than that of previous analysis ($n = 109$), and replaced the previous Oncoprint in the original Figure 3B. We also remade the CNV driver frequency bar plots in the previous Supplemental Figure 3B (new Supplemental Figure 5B). We additionally changed the Oncoprint in Figure 2A and the Kaplan-Meier curves in Figure 2B. For the copy number state assignment of a gene per sample, we used SNP6 arrays for 105 tumors and exome-EXCAVATOR for the remaining 4 tumors for which the SNP6 array was unable to be performed due to limited sample amount. We also modified the description in the section, “Driver gene mutations in CS” in the Results accordingly.

By ABSOLUTE (Carter *et al.* 2012) analysis of the SNP6 results ($n = 105$), the tumor ploidy was computed and the result is additionally shown in Figure 1. The high tumor ploidy was tightly associated with CNH subtype ($p = 0.0002$ by Mann Whitney U -test). For the remaining 4 tumor samples, FACETS (Shen and Seshan 2016) was used to compute tumor ploidy based on copy number information derived from exome-EXCAVATOR data. We added a description in the section, “Identification of four distinct genomic subtypes in CS samples” in the Results.

3. A REMARK diagram is essential: 169 samples were originally identified, of which 138 passed pathology review. 117 passed DNA quality control and genomic sequence data from 109 are presented. Only 100 underwent RNA-seq. These data are buried in the supplementary methods and should be readily visible to the reader with an explanation for why samples were discarded at various stages of the process.

We have prepared a REMARK diagram to integrate into one place the scattered information for the number of samples and the analytical results, and this is included as the new Supplemental Figure 2.

4. I am curious that a POLE-mutated ovarian sample was identified. The rate of POLE mutations in ovarian cancer is very low (2 validated in the TCGA cohort for example) and I wonder whether this sample might have been a misclassified endometrial primary.

In view of this comment, we re-checked the medical record of the ovarian POLE case (OV594). The ovarian tumor was found to be associated with an endometriotic lesion in the ovary. Moreover, the grade-1 endometrioid carcinoma also existed in the uterine corpus. The endometrial tumor was histologically distinct from the carcinoma component (grade-2 endometrioid) of the ovarian CS. Being consistent with previously proposed criteria, which include the presence of ovarian endometriosis (Scully *et al.* 1998), this case was diagnosed as synchronous endometrial and ovarian cancer (SEOC).

Based on exome sequencing analyses of the ovarian CS and the synchronous endometrial carcinoma, we found that a fraction of SNVs/indels including *POLE* p. P286R mutation were shared by both tumors. This observation is indeed consistent with the findings of previous NGS analyses, in which most pairs of primary tumors from SEOC patients shared relevant driver events and, therefore, most of the synchronous primaries were clonally derived (Anglesio *et al.* 2016; Schultheis *et al.* 2016). Together with the association of endometriosis for the ovarian CS, we concluded that the ovarian *POLE* CS of OV594 was likely to have originated from the endometria, sharing ancestry with the synchronous endometrial carcinoma.

Further investigation of the medical records for all cases in the cohort identified an additional case (OV343) as SEOC with ovarian endometriosis. We added the brief description for this condition in the section, “Identification of four distinct genomic subtypes in CS samples” in the Results, and in the section, “Histopathological diagnosis” in the Supplemental Materials and Methods. We included the details of these changes as Supplemental Table 1B and Supplemental Figure 4.

5. Iniparib is no longer considered to be a PARP inhibitor, so the relevant comment (end of page 11, start of page 12) should be removed. Moreover, as a result, the negative trial of iniparib (ref 64) in uterine CS proves nothing about whether or not defective HR is a meaningful therapeutic target in CS. This whole sentence should be removed.

At the time of the initial submission, we did not realize that iniparib was no longer regarded as a PARP inhibitor. We have therefore removed the description for iniparib in the Discussion and instead added a description regarding a case report in which olaparib showed efficacy in a patient with a germline *RAD51D*-mutated CS (Chandran and Kennedy 2018).

Reviewer #2 (Remarks to the Author)

This descriptive paper presents data from an exhaustive genomic analysis of CS with the aim of resolving its origin and key driver genes. The paper succeeds in convincingly clarifying that CS are not collision tumours although this had partly been shown in previous studies. The other significant finding is that CS cancer fall into 4 molecular subtypes that are reflective of those already described for endometrial cancers analysed by the TCGA. It is interesting that the sarcoma component of the tumours are associated with expression of genes known to be involved in EMT. Overall this paper provides some interesting insight into this sub-type of endometrial/ovarian cancers, although the clinical implications of this work remain unclear and not really addressed by the authors.

1. A limitation of the study is the use of only a targeted sequencing panel which has been useful in clonality analysis and understanding the role of known driver genes but does not represent a defining study of all driver genes. Related to this is the lack of clarity as to how the sequencing, expression and methylome data was generated. Maybe I missed it, but the only description I could find was the targeted and exome sequencing platform but nothing about the expression and methylation. In addition, how many of the cases were FFPE and fresh frozen. There is no mention of access to matching normal DNA so how were somatic mutations distinguished from germline variants. These details are required in order to assess the robustness of the actual data and subsequent interpretations.

Per the comment regarding the limitation of target panel sequencing in the identification of possible driver genes, we have included a sentence in the Discussion to address this comment:

“...; the current study, however, relies mostly on targeted panel sequencing, which limits the identification of potential novel drivers.”

We agree with Reviewer #2 (and also with the Reviewer #3) that our previous descriptions for the methodologies, including our informatics tools, were short and insufficient to some extent. We further noted some difficulty in being able to easily identify the full descriptions for the samples and the methods in the Supplemental Materials and Methods and, particularly from the Main Document. Although we did

mention the use of matched normal DNA for all of the target panel and exome sequencing analyses, this information might have also been buried in the previous descriptions in our Materials and Methods sections.

To rectify this, first, we prepared new Supplemental Tables to describe the samples used in the multi-regional sequencing and the matched normal DNA sources, and assembled the old and new tables of the sample descriptions into the new Supplemental Tables 2A, 2B, 2C, and 2D. The tables also have information pertaining to sample preservation type: fresh frozen (FF) or formalin-fixed paraffin-embedded (FFPE).

Second, in the Supplemental Materials and Methods, we have included more details to the sections of “Separate sampling and spatial multi-sampling of Carcinoma and Sarcoma Components” and “Sample preparation for genomic and epigenetic analysis”, and changed the section titles to “Processing tumor samples” and “DNA/RNA preparation for genomic and epigenetic analysis”, respectively. We also have included a section labeled as “Matched normal samples” for reader clarity.

Third, we have included a more detailed description of the analysis in the section titled, “Bioinformatics tools to analyze sequencing data”, and have divided the section into 13 subsections for clarification, changing the overall section title to “Bioinformatics and statistics”.

Finally, to connect the Materials and Methods in the Main Document with those in the Supplemental Document, we inserted a section called “Histopathological diagnosis and processing tumor and matched normal samples”, and changed the section title of “Statistical analyses” to “Bioinformatics and statistical analyses”.

2. Related to the point above, in Figure 2, how were the somatic and germline mutations identified.

As mentioned above in the Response to Comment #1 from this Reviewer, we prepared a new section called “Bioinformatics and statistics” and have included more detail

regarding the bioinformatics tools in the Supplemental Materials and Methods. The methods to call germline and somatic variants and the annotations are now properly described in the subsections of “Germline variant call and variant pathogenicity interpretation” and “Somatic variant call”.

3. The authors present data showing that the sarcoma components of the CS tumours show characteristic features of EMT, which is in fact quite expected as they are mesenchymal. But are the authors suggesting the methylation of MIR200 *etc* are driver events? I suspect not but this needs to be stated explicitly. It would seem logical that some other influence is driving EMT and the gene expression changes are simply reflecting that new induced state. The authors did do whole exome sequencing on purified epithelial and sarcoma components of 4 CS tumors but do not report if there were any sarcoma specific alterations that might explain the EMT. Even if none were found this should be stated.

As the Reviewer has pointed out, we did not clearly mention our thoughts on the role of miR-200a/200b/429 and/or miR-141/200c during CS histogenesis in the previous version of the Discussion. We believe that our findings regarding hypermethylation of the promoter region supports a possible definitive role of the microRNAs in CS-cell sarcoma differentiation, which was proposed in the previous literature (Cherniack *et al.* 2017; Li *et al.* 2017). However, there may also be other mechanisms driving EMT. We have improved these comments in the new version of the Discussion. We have also added in the Discussion a description regarding the observed inverse relationship between the *CTNNB1* activating mutation and the miRNA promoter methylation in the current cohort, thereby highlighting the complex nature of EMT in CS.

Following the Reviewer’s suggestion, we re-assessed the results derived from the exome analyses of 6 tumor samples as pairs of selective carcinoma and sarcoma sampling, and 3 tumors as sets of multi-regional sampling with differential composition of carcinoma and sarcoma. No recurrent mutation in the SNVs/indels or CNVs was detected differentially in the carcinoma or sarcoma component among the 9 CS tumors. To include this negative result, we created a new section called “Genetic Events Possibly Promote Differentiation between Carcinoma and Sarcoma” in the Supplemental Document. We also modified the

associated descriptions in the sections of “Evolutionary analysis of carcinoma and sarcoma components with differential sampling” and “Genetic similarity of multiple regions with variable composition of carcinoma and sarcoma elements in a tumor” in the Results.

4. The authors should state that CS were previously known as a malignant mixed mullerian tumor. This provides some historical context.

We have inserted a description of “malignant mixed Mullerian tumor” in the Introduction.

5. Minor spelling/ grammatical errors.

- a. Page 6 there is an “of” missing; alteration-independent OF carcinoma

An “of” was not needed, as we intended for this to explain the type of differentiation as “independent of any alteration”. To avoid misleading readers, we changed the expression of “genomic alteration-independent carcinoma and sarcoma differentiation” to “... differentiation of the carcinoma and sarcoma components is independent of genomic alterations”.

- b. Page 7. microsatellite instable (MSI) should be microsatellite instability.

- c. Page 16. "These results suggest less biological significance of the driver... should be BIOLOGICAL.

- d. Page 17. each tumor samples represent one genomic. should be SAMPLE

We corrected these wrong terms/typos accordingly.

Reviewer #3 (Remarks to the Author)

In this manuscript entitled “Clinically-relevant Molecular Subtypes and Genomic Alteration-independent Differentiation in Gynecologic Carcinosarcoma” by Gotoh *et al.*, the authors performed multi-omics data analyses for a relatively large dataset of carcinosarcoma and clarified four molecular subtypes based on genomic alterations. They also demonstrated that epithelial mesenchymal transition (EMT) was a key mechanism of sarcoma differentiation by using transcriptome and DNA methylome analyses. In addition, comparison of genomic/transcriptomic alterations between their dataset and TCGA dataset is highly evaluated in terms of enhancement of this study’s quality. However, they did not show the detail of this study in the manuscript including supplementary information. The reproducibility and reliability of the results are most important in bioinformatics field. In particular, they should describe “Materials and Methods” in much more detail. Detailed comments are described below.

In response to the comment regarding the need to include more details about materials and methods, we have made maximal efforts to add more precise and detailed information into the Supplemental Materials and Methods. For bioinformatics methodologies in particular, we have added the section, “Bioinformatics and statistics,” which comprises 13 subsections, each of which describe the analytical method in detail.

1. Abstract (Page 2): Carcinosarcoma of the uterus is not so rare compared to of the ovary.

We have changed the expression of “rare” to “with a low incidence rate” in the Abstract.

2. Abstract (Page 2, Lines 6-7): The authors described as “Panel sequencing of 109 CS with 596 genes” in the abstract. However, in Supplementary Materials and Methods, they described as “105 and 4 cases for CS panels and exome, respectively”.

We changed the expression of “Panel sequencing of 109 CS with 596 genes” to “Sequencing 109 CS samples with focusing on 596 genes, ...”.

3. Abstract (Page 2, Line 8): The authors should avoid using non-recognized abbreviations such as “CNH and CNL”.

As suggested by the Reviewer, we have changed the words from “CNH” and “CNL” to “copy number high” and “copy number low”, respectively.

4. Abstract (Page 2, Line 13): This sentence (“The current study thus provides novel therapeutic options for CS”) is apparently overstated. The authors should conclude based on their results.

Following this suggestion, we replaced the word of “options” with “possibilities”.

5. Introduction (Page 5, Lines 6-8): TCGA pan-cancer study demonstrates that highly enriched events are primarily mutations in uterine corpus endometrial cancer (PMID: 24071851).

We have added another reference that describes a TCGA pan-cancer study (Ciriello *et al.* 2013) into the sentence.

6. Results (Page 7, Lines 3-4) and Supplementary Materials and Methods (Suppl. M&M) (Page 16): How many samples were used in the molecular subtyping? Did the authors use both targeted and exome sequencing data in this analysis? If so, how did you merge two types of data into one? Data quality of targeted genes are different between targeted sequencing data and exome sequencing data.

For our initial submission, we used 105 target panel samples as well as 4 exome samples for genomic aberration subtyping. As stated in the comment, we did not clearly mention how we merged the data in the previous Supplemental Materials and Methods. We have addressed this concern by providing information regarding how we merged the files in the section labeled “Library preparation and sequencing analysis”, and in more detail in the subsection “Somatic variant call” under “Bioinformatics and statistics” in the new Supplemental Materials and Methods. In terms of data quality, we show a median depth of target panels and exome sequencing in the section, “Library preparation and sequencing analysis”. Note that in this revision, we performed exome analyses for OV594 in the response to the Comment #4 by Reviewer #1, changing the number of target panel data and exome data for CS to 104 and 5 samples, respectively.

7. Results (Page 7) and Suppl. M&M (Page 16): How did the authors determine MSI-high status?

MSI-high status (microsatellite instability; MSI) was determined by deviations from paired normal control samples in electropherograms of two or more among 6 DNA markers (BAT25, BAT26, D2S123, D5S346, D17S250 and BAT40), as previously described (Boland *et al.* 1998). Following the Reviewer's comment, we have added a description of how we determined MSI-high status in the subsection, "Molecular subtyping based on genomic aberration profiles" of the "Bioinformatics and statistics" section in the Supplemental Materials and Methods.

8. Results (Page 7) and Suppl. M&M (Page 14): Several methods (including EXCAVATOR) to detect copy number alterations from exome sequencing data have a limitation to infer copy number across whole genome because of no use of sequencing reads from intergenic or intronic regions. On the other hand, copy number alterations is one of the most important information in this study. Therefore, the authors should validate copy number alterations by other modalities.

The comment pertaining to CNV analysis is related to Comment #2 made by Reviewer #1. In response to these comments, we additionally performed SNP6 arrays with 105 CS samples for which the DNA remained available, as a standard method to detect somatic CNVs in tumor samples.

For each sample, the copy number status for each gene on the panel (estimated with high-level thresholds in GISTIC analysis) was mostly concordant between SNP6 array outputs and EXCAVATOR results derived from targeted resequencing data; the percentage of CNV call concordance was 98.04% (median; range, 69.82%–100%) for the 105 samples. As such, the SNP6 assays validated the previous CNV calls obtained through the panel sequencing–EXCAVATOR analysis. We added a brief description regarding the concordance in the section "Somatic variant call" in the Supplemental Materials and Methods.

Nevertheless, in appreciation of the high resolution and wide coverage of the SNP6 data, following the suggestions by this Reviewer and Reviewer #1, we conducted two additional analyses based on the SNP6 CNV information to better characterize CS tumors.

GISTIC analysis of the SNP6 copy number data changed the results of the driver CNV calling from that determined using the panel sequencing data. We made an Oncoprint figure for this result with the slightly smaller sample size ($n = 105$) than the previous analysis ($n = 109$) and replaced the previous Oncoprint figure in Figure 3B. We also remade the CNV driver frequency bar plots depicted in the previous Supplemental Figure 3B (new Supplemental Figure 5B). We also modified the description in the section “Driver gene mutations in CS” in the Results accordingly.

Tumor ploidy was computed by ABSOLUTE analysis based on the SNP6 results (Carter *et al.* 2012), and this is shown in part of Figure 1. We added a description in “Identification of four distinct genomic subtypes in CS samples” in the Results.

9. Results (Page 9, Lines 4-5): Although the authors used data derived from 57 TCGA uterine CS, they did not describe the detail of TCGA data in the manuscript as well as Suppl. M&M. Which type of data did they use? Did they download FASTQ or BAM files from GDC and detect mutations and copy number alterations by themselves?

We have added a description of the analytical method used to process the TCGA data in the subsection “TCGA data analyses” in the “Bioinformatics and statistics” section in the Supplemental Materials and Methods.

10. Results (Page 12, Lines 14-15): The authors should describe a proper reference.

We added an appropriate reference.

11. Results (Page 13, Lines 3): Again, the authors should avoid using a non-recognized abbreviation “JFCR” before any explanation.

As suggested by the Reviewer, we provide a full term of the “Japanese Foundation for Cancer Research” for “JFCR”.

12. Results (Page 13): Again, the authors did not describe the detail of TCGA data in the manuscript as well as Suppl. M&M.

As mentioned above for Comment #9, we have added a description of the analytical method used to process the TCGA data in the subsection “TCGA data analyses” in the “Bioinformatics and statistics” section in the Supplemental Materials and Methods.

13. Results (Page15) and Figure 5: The authors demonstrated phylogenetic trees by using 596-gene panels. However, CNH samples harbor little SNV/indels and estimation of clonal evolution based on SNV/indels is not suitable for CNH samples. They could perform estimation of clonal evolution based on copy number alterations if they succeeded to validate copy number alterations by other modalities. And they also could show similarities or differences in mutation and copy number alteration profiles between carcinoma and sarcoma components in the same tumor as a supplementary table. This information might be useful when they will discuss about therapeutic strategies for CS.

As mentioned above, we confirmed concordance in the copy number status of each gene on the panel between SNP6 array outputs and the targeted resequencing–EXCAVATOR results. In response to this comment, we performed phylogenetic tree analyses with TuMult (Letouze *et al.* 2010) based on CNV information derived from sequenced data to assess reproducibility of the tree analyses using the SNVs/indels of the 596 genes. Among 14 tumors with selective sampling of carcinoma and sarcoma components, data for six tumors were acquired with exome sequencing. Therefore, for these 6 tumors, we also estimated the clonal evolution processes with SNVs/indels and with CNVs using exome data (Supplemental Figure 9). These analyses confirmed the pattern of branching, with

generally longer trunks and shorter branches for carcinoma and sarcoma components in a CS tumor, including those in the CNH subtype (Supplemental Figure 9).

Further in response to this comment, we have prepared new Supplemental Tables 4B and 4C for SNVs/indels with the mutant allele frequency and CNVs with copy number, respectively, to provide detailed mutational information of the 596 genes for selectively sampled carcinoma and sarcoma elements in a tumor. We show copy number information in the limited region, which was included in both the 596-gene panel and in regions identified as driver segments by GISTIC analysis of bulk tumor SNP6 data (Figure 3B). We have amended our description in the section “Evolutionary analysis of carcinoma and sarcoma components with differential sampling” in the Results. We have also added a description in the Discussion regarding a therapeutic strategy that is based on driver events shared by the carcinoma and sarcoma components in the tumor.

14. Results (Page 17) and Suppl. M&M (Page 9): How did the authors determine the percentages of carcinoma and sarcoma components? How about tumor purity?

The proportion of carcinoma and sarcoma elements in a region was determined by a pathologist based on the number of tumor nuclei. We have included a description in the section “Processing tumor samples” in the Materials and Methods.

Tumor purity was estimated using B-allele frequency in a loss of heterozygosity (LOH) region using ExomeCNV (Sathirapongsasuti *et al.* 2011). Tumor purity is additionally shown as a color map in Figure 1.

15. Materials and Methods (Page 24) and Suppl. M&M (Page 12): Why did the authors use two types of SureSelect (v4 or v5)? They should clarify which data was based on SureSelect v4 or v5.

For library preparations, we switched the hybridization capture kit from the SureSelect Human All Exon V4 to the V5 (Agilent Technologies) in January 2013. Therefore, the V4

and V5 kits were used before and after January 2013, respectively. We described this version change in the Materials and Methods, and indicate which kit was used for each sample in Supplemental Tables 2A, 2B, and 2C.

16. Suppl. M&M (Page 9): The authors should describe tumor purity in each sample used in genomic analyses. Genomic data based on bulk sample is influenced by tumor purity.

Tumor purity was estimated using B-allele frequency in a loss of heterozygosity (LOH) region using ExomeCNV (Sathirapongsasuti *et al.* 2011). Tumor purity is additionally shown as a color map in Figure 1.

17. Suppl. M&M (Page 10): The authors should describe which type of normal tissue was used. And they should also describe the information of matched normal sample per case. Selection of normal sample is very important for copy number detection.

We have now included a table (Supplemental Table 2D) to describe the source of normal DNA per case.

18. Suppl. M&M (Page 12): It is well known that RPKM is not good to compare across samples. It may be better to select Transcripts Per Kilobase Million (TPM) compared to RPKM or FPKM.

As implied by the Reviewer, our results may change if we switch the expression metrics from FPKM to TPM. To determine the effect of a switch on the clustering analysis, relying on the TPM output from RSEM-1.3.0 (Li and Dewey 2011), we repeated the consensus clustering, which had been originally performed with FPKM data in the previous Supplemental Figure 5 (new Supplemental Figure 7). We then compared the TPM and FPKM results. The analysis provided a highly concordant cluster assignment between the TPM and FPKM formats (96/97 samples; 99.0%), indicating a consistency in the data between TPM and FPKM formats for this clustering analysis.

Next, we investigated the effect of using TPM on gene expression signature analysis.

Making use of the TPM data derived from TCGA-UCEC RNA-seq, we extracted the gene expression signature to distinguish uterine endometrioid from serous histologies (herein designated as ES signature), which had also been originally developed with the FPKM-format data in Figure 4. The resultant signatures with a cut-off q -value < 0.05 and AUC > 0.8 in the SAM-ROC analysis showed considerable overlap ($> 60\%$) in the gene components between the two methods, as shown in Table R1.

Table R1: Number of gene components in the uterine ES signatures derived from TPM and FPKM data of TCGA-UCEC RNA-seq.

Signature	TPM only	Shared by TPM and FPKM	FPKM only	Total
Endometrioid	1 (0.8%)	76 (64.4%)	41 (34.7%)	118 (100%)
Serous	10 (6.4%)	134 (86.4%)	11 (7.1%)	155 (100%)

Importantly, even though many of the signature genes were extracted differently between the two formats, the first principal component of the ES signature from the TPM analysis exhibited a strong correlation with that from the FPKM analysis (TCGA-UCEC; Spearman $Rho = 0.99830$ and $p < 0.00001$, JFCR-CS; Spearman $Rho = 0.99565$ and $p < 0.00001$). The heatmaps of the ES signature were also indistinguishable between the two formats.

As such, we observed no clear difference between TPM and FPKM in our analyses. Therefore, we concluded that there was no need to switch the expression metrics from FPKM to TPM in this study.

Abbreviations: TPM; transcripts per kilobase million, FPKM; fragments per kilobase million, SAM; significance analysis of microarrays, ROC; receiver operating characteristic curve, TCGA; the Cancer Genome Atlas, UCEC; uterine corpus endometrial cancer, JFCR; Japanese Foundation for Cancer Research and CS; carcinosarcoma.

References

- Anglesio MS, Wang YK, Maassen M, Horlings HM, Bashashati A, Senz J, Mackenzie R, Grewal DS, Li-Chang H, Karnezis AN *et al.* 2016. Synchronous Endometrial and Ovarian Carcinomas: Evidence of Clonality. *J Natl Cancer Inst* **108**: djv428.
- Boland CR, Thibodeau SN, Hamilton SR, Sidransky D, Eshleman JR, Burt RW, Meltzer SJ, Rodriguez-Bigas MA, Fodde R, Ranzani GN *et al.* 1998. A National Cancer Institute Workshop on Microsatellite Instability for cancer detection and familial predisposition: development of international criteria for the determination of microsatellite instability in colorectal cancer. *Cancer Res* **58**: 5248-5257.
- Cancer Genome Atlas Research N. 2011. Integrated genomic analyses of ovarian carcinoma. *Nature* **474**: 609-615.
- Carter SL, Cibulskis K, Helman E, McKenna A, Shen H, Zack T, Laird PW, Onofrio RC, Winckler W, Weir BA *et al.* 2012. Absolute quantification of somatic DNA alterations in human cancer. *Nature biotechnology* **30**: 413-421.
- Chandran EA, Kennedy I. 2018. Significant Tumor Response to the Poly (ADP-ribose) Polymerase Inhibitor Olaparib in Heavily Pretreated Patient With Ovarian Carcinosarcoma Harboring a Germline RAD51D Mutation. *JCO Precision Oncology* doi:10.1200/po.18.00253: 1-4.
- Cherniack AD, Shen H, Walter V, Stewart C, Murray BA, Bowlby R, Hu X, Ling S, Soslow RA, Broaddus RR *et al.* 2017. Integrated Molecular Characterization of Uterine Carcinosarcoma. *Cancer Cell* **31**: 411-423.
- Ciriello G, Miller ML, Aksoy BA, Senbabaoglu Y, Schultz N, Sander C. 2013. Emerging landscape of oncogenic signatures across human cancers. *Nature genetics* **45**: 1127-1133.
- Letouze E, Allory Y, Bollet MA, Radvanyi F, Guyon F. 2010. Analysis of the copy number profiles of several tumor samples from the same patient reveals the successive steps in tumorigenesis. *Genome biology* **11**: R76.
- Li B, Dewey CN. 2011. RSEM: accurate transcript quantification from RNA-Seq data with or without a reference genome. *BMC bioinformatics* **12**: 323.
- Li J, Xing X, Li D, Zhang B, Mutch DG, Hagemann IS, Wang T. 2017. Whole-Genome DNA Methylation Profiling Identifies Epigenetic Signatures of Uterine Carcinosarcoma. *Neoplasia* **19**: 100-111.

- Sathirapongsasuti JF, Lee H, Horst BA, Brunner G, Cochran AJ, Binder S, Quackenbush J, Nelson SF. 2011. Exome sequencing-based copy-number variation and loss of heterozygosity detection: ExomeCNV. *Bioinformatics* **27**: 2648-2654.
- Schultheis AM, Ng CK, De Filippo MR, Piscuoglio S, Macedo GS, Gatus S, Perez Mies B, Soslow RA, Lim RS, Viale A *et al.* 2016. Massively Parallel Sequencing-Based Clonality Analysis of Synchronous Endometrioid Endometrial and Ovarian Carcinomas. *J Natl Cancer Inst* **108**: djv427.
- Scully RE, Young RH, Pathology AFIO, Clement PB, Research UAf, Pathology Ei. 1998. *Tumors of the Ovary, Maldeveloped Gonads, Fallopian Tube, and Broad Ligament*. Armed Forces Institute of Pathology under the auspices of Universities Associated.
- Shen R, Seshan VE. 2016. FACETS: allele-specific copy number and clonal heterogeneity analysis tool for high-throughput DNA sequencing. *Nucleic acids research* **44**: e131.

REVIEWERS' COMMENTS:

Reviewer #1 (Remarks to the Author):

In this revision, the authors have re-written the manuscript extensively in response to reviewer comments.

My original review had two main comments

1. Use of an endometrial cancer classifier to analyse ovarian cancer samples. The authors have partially addressed this by separating the ovarian samples from the endometrial in Figure 4 and undertaken an analysis relative to the ovarian TCGA dataset. As one had suspected, this highlighted that the ovarian cancer and endometrial carcinosarcomas are in some respects very different from each other. The discussion section should be amended to include a broader discussion on how the ovarian and endometrial samples differ.
2. Copy number analyses. Both other reviewers also commented that gene panels were suboptimal for analysis of copy number aberrations, and so the authors have undertaken SNP6 array analyses on 105 samples, and I think that this underscores the extent of the CNA in the ovarian and copy number high endometrial samples.

Two other comments

1. EMT gene expression signatures. I note that the authors use the EMT signature described by Byers et al (reference 54). There are multiple EMT signatures available – why was this one selected and do the authors derive the same result with another signature e.g. Mak et al Clin. Cancer Res. 2016 or Tan et al EMBO Mol. Med. 2014?
2. I notice that the Supplemental Figures 12 and 13, which relate to immune microenvironment associations in carcinosarcoma, are not mentioned anywhere in the results section. Either they should be specifically mentioned, or they should be removed. I would favour the former as there appear to be some interesting data in those two figures.

Reviewer #2 (Remarks to the Author):

The authors have made extensive changes to the manuscript in response to the reviewers comments and suggestions. The inclusion of the SNP6 data represents a significant improvement on the data from the targeted sequencing. Overall the authors have adequately addressed the issues raised by the reviewers.

Reviewer #3 (Remarks to the Author):

The authors have responded to all comments from the reviewers. The current version is suitable for Nature Communications.

Reviewer #1 (Remarks to the Author):

In this revision, the authors have re-written the manuscript extensively in response to reviewer comments.

My original review had two main comments

1. Use of an endometrial cancer classifier to analyse ovarian cancer samples. The authors have partially addressed this by separating the ovarian samples from the endometrial in Figure 4 and undertaken an analysis relative to the ovarian TCGA dataset. As one had suspected, this highlighted that the ovarian cancer and endometrial carcinosarcomas are in some respects very different from each other. The discussion section should be amended to include a broader discussion on how the ovarian and endometrial samples differ.

Following this suggestion, we added a description regarding the difference of ovarian and endometrial CSs in the Discussion.

2. Copy number analyses. Both other reviewers also commented that gene panels were suboptimal for analysis of copy number aberrations, and so the authors have undertaken SNP6 array analyses on 105 samples, and I think that this underscores the extent of the CNA in the ovarian and copy number high endometrial samples.

Two other comments

1. EMT gene expression signatures. I note that the authors use the EMT signature described by Byers *et al.* (reference 54). There are multiple EMT signatures available – why was this one selected and do the authors derive the same result with another signature e.g. Mak *et al* Clin. Cancer Res. 2016 or Tan *et al* EMBO Mol. Med. 2014?

We used the EMT score developed by Byers *et al.*¹, because the score was used in TCGA uterine carcinosarcoma study². As suggested by the Reviewer #1, we compared the Byer EMT score with those by Mak *et al.*³ and by Tan *et al.*⁴ with Spearman correlation analyses of 97 CS RNA-seq data. For simplicity, we used the first principal component of each of three different EMT signatures. The Byer EMT score indeed exhibited strong correlations with those by Mak and Tan ($Rho = 0.7999$ and 0.9360 ; $p < 0.0001$ and $p < 0.0001$, respectively), indicating robustness of the score. We added a brief description in the section, “EMT score and micro-environmental analysis” in the Methods.

2. I notice that the Supplemental Figures 12 and 13, which relate to immune microenvironment

associations in carcinosarcoma, are not mentioned anywhere in the results section. Either they should be specifically mentioned, or they should be removed. I would favour the former as there appear to be some interested data in those two figures.

Whereas we appreciate the Reviewer's interest in this topic regarding immune microenvironment, we also realize the current manuscript became considerably long and the topic was largely independent of the other parts in the study. Following this suggestion, we decided to remove this topic; the section "Immune Microenvironment of CS" in the previous Supplemental Text, and Supplemental Figures 12 and 13, from the final version of the manuscript.

Reviewer #2 (Remarks to the Author):

The authors have made extensive changes to the manuscript in response to the reviewers comments and suggestions. The inclusion of the SNP6 data represents a significant improvement on the data from the targeted sequencing. Overall the authors have adequately addressed the issues raised by the reviewers.

Reviewer #3 (Remarks to the Author):

The authors have responded to all comments from the reviewers. The current version is suitable for Nature Communications.

References

1. Byers LA, *et al.* An epithelial-mesenchymal transition gene signature predicts resistance to EGFR and PI3K inhibitors and identifies Axl as a therapeutic target for overcoming EGFR inhibitor resistance. *Clin Cancer Res* **19**, 279-290 (2013).
2. Cherniack AD, *et al.* Integrated Molecular Characterization of Uterine Carcinosarcoma. *Cancer Cell* **31**, 411-423 (2017).
3. Mak MP, *et al.* A Patient-Derived, Pan-Cancer EMT Signature Identifies Global Molecular Alterations and Immune Target Enrichment Following Epithelial-to-Mesenchymal Transition. *Clin Cancer Res* **22**, 609-620 (2016).
4. Tan TZ, *et al.* Epithelial-mesenchymal transition spectrum quantification and its efficacy in deciphering survival and drug responses of cancer patients. *EMBO Mol Med* **6**, 1279-1293 (2014).